# Surplus fatty acid synthesis increases oxidative stress in adipocytes and induces lipodystrophy

Li Weng[1,2,11], Wen-Shuai Tang[1,11], Xu Wang[3,11], Yingyun Gong [4,11], Changqin Liu [5], Ni-Na Hong[2], Ying Tao[1], Kuang-Zheng Li[1], Shu-Ning Liu[6], Wanzi Jiang[4], Ying Li[7], Ke Yao [8], Li Chen [1], He Huang [1], Yu-Zheng Zhao [6], Ze-Ping Hu [8], Youli Lu[9], Haobin Ye[1], Xingrong Du[1], Hongwen Zhou [4] ✉, Peng Li [1,10] ✉ & Tong-Jin Zhao [1,10] ✉

Adipocytes are the primary sites for fatty acid storage, but the synthesis rate of fatty acids is very low. The physiological significance of this phenomenon remains unclear. Here, we show that surplus fatty acid synthesis in adipocytes induces necroptosis and lipodystrophy. Transcriptional activation of FASN elevates fatty acid synthesis, but decreases NADPH level and increases ROS production, which ultimately leads to adipocyte necroptosis. We identify MED20, a subunit of the Mediator complex, as a negative regulator of FASN transcription. Adipocyte-specific male *Med20* knockout mice progressively develop lipodystrophy, which is reversed by scavenging ROS. Further, in a murine model of HIV-associated lipodystrophy and a human patient with acquired lipodystrophy, ROS neutralization significantly improves metabolic disorders, indicating a causal role of ROS in disease onset. Our study well explains the low fatty acid synthesis rate in adipocytes, and sheds light on the management of acquired lipodystrophy.

Adipose tissue is the primary site for lipid storage and an important endocrine organ. It plays a pivotal role in maintaining the whole-body metabolic homeostasis[1]. Despite that adipose tissue stores large amounts of fatty acids, it has been well appreciated that adipose tissue itself does not synthesize much fatty acids[2–4]. The majority of the stored fatty acids are from dietary fat or made by the liver. However, the physiological significance of the low de novo fatty acid synthesis rate in adipocytes remains unclear.

[1]State Key Laboratory of Genetic Engineering, Shanghai Key Laboratory of Metabolic Remodeling and Health, Institute of Metabolism and Integrative Biology, Drug Clinical Trial Center, Shanghai Xuhui Central Hospital / Zhongshan-Xuhui Hospital, Zhongshan Hospital, Fudan University, Shanghai, China. [2]State Key Laboratory of Cellular Stress Biology, School of Life Sciences, Xiamen University, Xiamen, China. [3]School of Life Science, Anhui Medical University, Research Center for Translational Medicine, the Second Affiliated Hospital of Anhui Medical University, Hefei, Anhui, China. [4]Department of Endocrinology and Metabolism, the First Affiliated Hospital of Nanjing Medical University, Nanjing, China. [5]Department of Endocrinology and Diabetes, the First Affiliated Hospital, Xiamen University, Xiamen, Fujian, China. [6]Optogenetics & Synthetic Biology Interdisciplinary Research Center, Shanghai Frontiers Science Center of Optogenetic Techniques for Cell Metabolism, State Key Laboratory of Bioreactor Engineering, East China University of Science and Technology, Shanghai, China. [7]Department of Endocrinology, Northern Jiangsu People's Hospital, Yangzhou, Jiangsu, China. [8]School of Pharmaceutical Sciences, Tsinghua-Peking Joint Center for Life Sciences, Beijing Frontier Research Center for Biological Structure, Tsinghua University, Beijing, China. [9]Shanghai Engineering Research Center of Phase I Clinical Research & Quality Consistency Evaluation for Drugs, Institute of Clinical Mass Spectrometry, Shanghai Academy of Experimental Medicine, Shanghai, China. [10]Tianjian Laboratory of Advanced Biomedical Sciences, School of life sciences, Zhengzhou University, Zhengzhou, Henan, China. [11]These authors contributed equally: Li Weng, Wen-Shuai Tang, Xu Wang, Yingyun Gong. ✉e-mail: drhongwenzhou@njmu.edu.cn; li-peng@mail.tsinghua.edu.cn; zhaotj@fudan.edu.cn

Excessive adipose tissue in obesity is closely associated with insulin resistance, fatty liver and dyslipidemia[5], whilst lack of functional adipose tissue causes a pathological condition called lipo-dystrophy, characterized by similar metabolic disorders[6]. Lipodystro-phies are rare heterogeneous disorders that are inherited or acquired[7,8]. While inherited lipodystrophies are largely due to genetic mutations[9,10], the causes of acquired lipodystrophies are more diverse[11]. The most prevalent form of acquired lipodystrophy occurs in HIV patients who underwent highly active antiretroviral therapy[9]. Other forms of acquired lipodystrophies are poorly diagnosed or studied. Due to the heterogenous nature of the disease, current treatment options are mainly aiming at the metabolic complications and have variable efficacy[9]. It is in urgent need to understand the underlying mechanism of acquired lipodystrophy and to develop more effective management strategies.

Oxidative stress is an essential factor in the pathogenesis of many diseases[12]. High level of reactive oxygen species (ROS) impairs adipo-genesis, induces insulin resistance and adipocyte hypertrophy, and causes metabolic disorders including type 2 diabetes and cardiovas-cular diseases[13,14]. Increased oxidative stress in adipocytes has also been observed in inherited or acquired lipodystrophies[15–17], but the causal relationship between them remains unclear.

Here, to explore the physiological significance of the low de novo fatty acid synthesis rate in adipocytes, we activated the transcription of fatty acid synthase (FASN) in adipocytes by the CRISPR-based activa-tion system. The de novo fatty acid synthesis rate indeed increased, but the adipocytes died of necroptosis due to increased ROS pro-duction. We identified MED20, a subunit of the Mediator complex, as a suppressor of *Fasn* transcription. We showed that knocking out *Med20* in adipocytes induced lipodystrophy in mice, which was largely reversed by either scavenging ROS or inhibiting necroptosis. Further-more, in a mouse model of HIV-associated lipodystrophy and in a human patient with acquired lipodystrophy, we demonstrated that increased ROS in adipocytes might be a driving force for acquired lipodystrophy. Taken together, our studies uncovered the physiolo-gical significance of the low fatty acid synthesis rate in adipocytes, and shed lights on the management of acquired lipodystrophy.

## Results
### Transcriptional activation of FASN in adipocytes leads to necroptosis
To address the physiological significance of the low fatty acid synthesis rate in adipocytes, we first examined the effect of artificially elevating fatty acid synthesis by overexpression of FASN in adipocytes. As the *Fasn* gene is fairly long (~7.5 kb) and hard to manipulate, we used the CRISPR-based activation system[18] to activate the endogenous tran-scription of *Fasn* (Fig. 1a). We obtained 3T3-L1 cells which showed a 2-fold increase in the mRNA and protein levels of FASN (Fig. 1b–d). We then differentiated the cells into adipocytes and treated them with uniformly [13]C-labeled glucose to evaluate de novo fatty acid synthesis (Fig. 1e). On the day of experiment, the control and FASN-overexpressing adipocytes did not show visible difference. As expec-ted, overexpression of FASN caused increased incorporation of [13]C into myristate (C14:0), palmitate (C16:0), palmitoleate (C16:1) and stearate (C18:0) (Fig. 1f). A close examination of [13]C incorporation into myr-istate and palmitate further illustrated the results (Fig. 1g and Sup-plementary Fig. 1a).

We went on to monitor the effect of excessive fatty acid synthesis on the morphology and survival of the adipocytes. On day 6 of dif-ferentiation, there was no visible difference between control and FASN-overexpressing adipocytes; however, on day 15, there was a 45% decrease in the cell number of the FASN-overexpressing adipocytes, as illustrated by the empty areas in the culture dish and increased LDH release into the culture medium (Fig. 1h–j). Consistent with increased fatty acid synthesis, the surviving FASN-overexpressing adipocytes had

a significantly higher percentage of big lipid droplets than control cells (Fig. 1k).

To confirm the results and rule out the off-target effect of the guide RNA of *Fasn*, we treated FASN-overexpressing adipocytes with an inhibitor of FASN, C75. As shown in Extended Data Fig. 1b–d, C75 treatment largely prevented cell death of FASN-overexpressing adi-pocytes and decreased the size of lipid droplets, confirming that overproduction of fatty acids causes cell death of the adipocytes.

We next sought to identify the type of cell death in FASN-overexpressing adipocytes by treating with different types of cell death inhibitors. The RIPK1 inhibitor NEC-1, but not the caspase inhi-bitor ZVAD or the ferroptosis inhibitor LIP-1, largely prevented cell death of FASN-overexpressing adipocytes (Fig. 1l–n), suggesting that the cells might die of necroptosis. To further strengthen the point, we checked the phosphorylated and total levels of necroptosis marker proteins, RIPK1, RIPK3 and MLKL[19,20]. The phosphorylated forms of RIPK1 (1.4-fold), RIPK3 (2.6-fold) and MLKL (3.3-fold) all significantly increased in FASN-overexpressing adipocytes (Fig. 1o; Supplemen-tary Fig. 1e).

Taken together, accelerated de novo fatty acid synthesis causes necroptosis of adipocytes.

### Increased ROS production drives necroptosis of FASN-overexpressing adipocytes
Next, we went on to identify the underlying mechanism of the necroptosis of FASN-overexpressing adipocytes. De novo fatty acid synthesis weighs heavily on the reducing agent NADPH. For each round of elongation (two carbons), it requires 2 molecules of NADPH. We therefore hypothesized that accelerated de novo fatty acid synthesis might decrease the cellular pool of NADPH, which could subsequently increase ROS production and cause cell death.

As fatty acid synthesis occurs in the cytosol, we monitored the cytosolic content of NADPH using the genetically encoded fluorescent iNAP sensor[21,22]. Figure 2a, b show that the NADPH level decreased by about 30% in FASN-overexpressing adipocytes. We also used a HYPER sensor[23] to monitor cellular content of $H_2O_2$, the major form of ROS. Figure 2c, d show that the $H_2O_2$ level increased about 2.3-fold in FASN-overexpressing adipocytes. To further confirm the results, we used CellROX, a specific fluorescent dye for ROS[24], to quantify the cellular ROS levels. Figure 2e, f show that the cellular ROS increased about 3.8-fold in FASN-overexpressing adipocytes. Thus, surplus synthesis of fatty acids indeed decreases NADPH level and increases ROS production.

To explore whether increased ROS production was the driving force of the cell death of FASN-overexpressing adipocytes, we treated the adipocytes with ROS scavenging reagents, BHA and GSH. As shown in Fig. 2g, h, both reagents could largely reverse the necroptosis of FASN-overexpressing adipocytes.

Taken together, overproduction of fatty acids causes necroptosis of the adipocytes by increasing ROS production.

### MED20 inhibits the transcription of *Fasn* through SNAIL and SLUG
We next sought to investigate the upstream regulators of *Fasn* tran-scription in adipocytes. In hepatocytes, SNAIL and SLUG have been reported to regulate the transcription of *Fasn*[25,26]. To examine whether SNAIL and SLUG might also control the transcription of *Fasn* in adi-pocytes, we knocked down either one or both of them. As shown in Fig. 3a, knockdown of either of them increased the mRNA level of *Fasn*. Furthermore, cell death of adipocytes was induced by knockdown of either SNAIL or SLUG, which was worsened by double knockdown of them (Fig. 3b, c; Supplementary Fig. 2a). In addition, it was necroptotic cell death induced by SNAIL and SLUG double knockdown in adipo-cytes, as it could be rescued by NEC-1 (Supplementary Fig. 2b–d). Furthermore, scavenging ROS with GSH also largely reversed cell

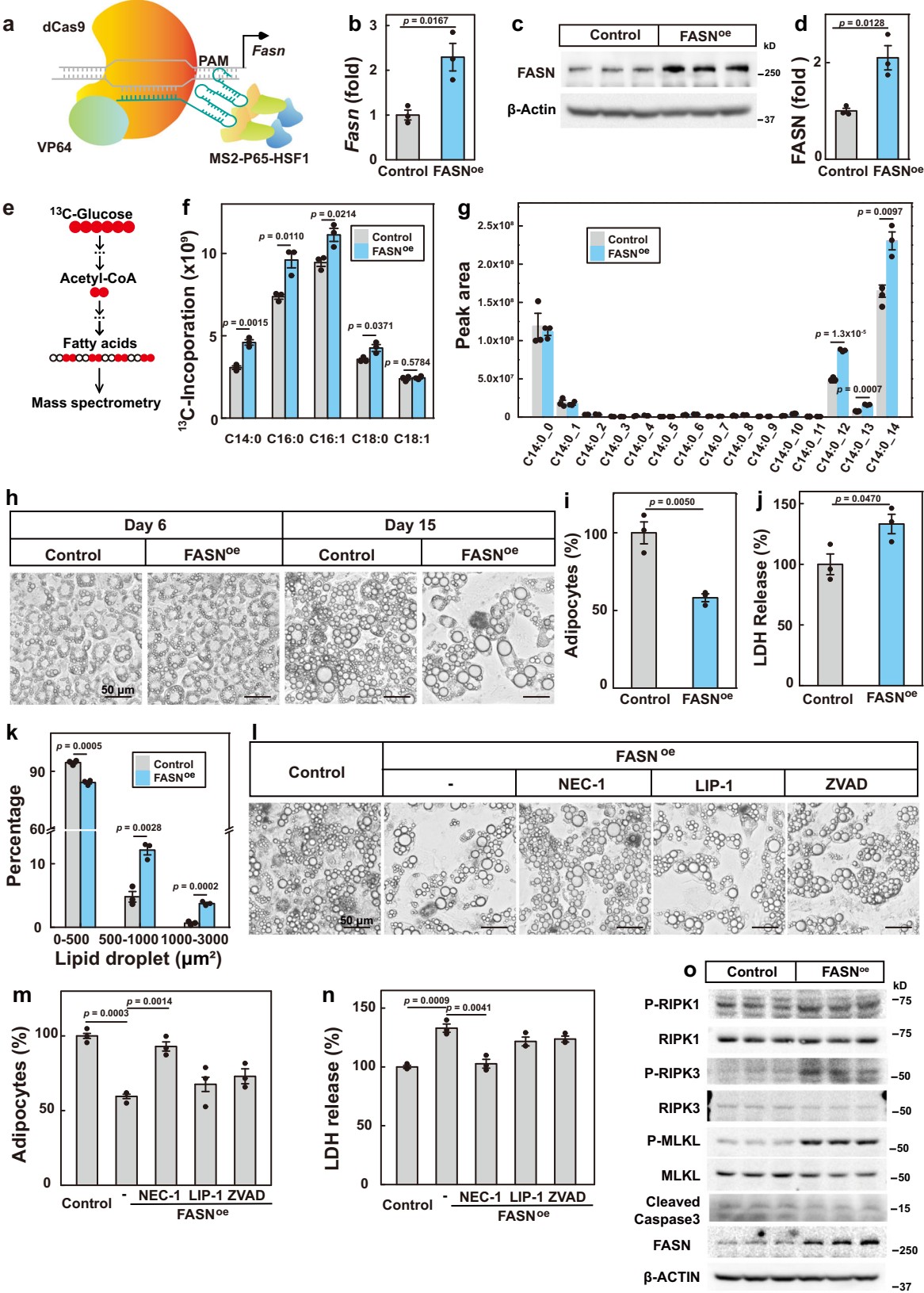

death (Fig. 3d, e; Supplementary Fig. 2e). Thus, SNAIL and SLUG are required for the survival of the adipocytes by inhibiting the transcription of *Fasn*.

We went on to identify the upstream regulators of SNAIL and SLUG. In preadipocytes, we have previously identified MED20, a non-essential subunit of the Mediator complex, as a key regulator of adipogenesis[27]. MED20 directly binds C/EBPβ and controls the transcription of the central regulator of adipogenesis, PPARγ and C/EBPα[27]. We re-visited the chromatin immunoprecipitation sequencing (ChIP-Seq) data performed in 3T3-L1 cells[27], and found that MED20 bound on the promoter regions of *Snai1* and *Snai2* (Fig. 3f), which encode SNAIL and SLUG, respectively. Furthermore,

**Fig. 1 | Transcriptional activation of FASN in adipocytes leads to necroptosis.**
**a** The strategy to transcriptionally activate FASN in 3T3-L1 cells using the CRISPR-based activation system. mRNA (**b**) and protein level (**c**) analysis of FASN in control and FASN-overexpressing (FASNᵒᵉ) adipocytes. The protein level was quantified in **d**. 36B4 and β-actin was used as an internal control, respectively. **e**–**g** On day 9 of differentiation, control and FASNᵒᵉ adipocytes were switched to medium containing uniformly $^{13}$C-labeled glucose (25 mM) for 24 h. On day 10, cells were harvested and subjected to de novo fatty acid synthesis analysis as described in *Methods*. **f** Incorporation of $^{13}$C into the indicated fatty acids were analyzed. **g** An illustration of the $^{13}$C incorporation into myristate (C14:0). On day 6 and day 15 of differentiation, cells were harvested for imaging under bright field (**h**). On day 15, the

number of adipocytes in each image (**i**), the released LDH in the medium (**j**) and size of lipid droplets (**k**) were analyzed. The number and size of lipoid droplet were quantified using Imaris 9.5. Starting from day 9 of differentiation, control and FASNᵒᵉ adipocytes were treated with NEC-1 (20 μM), ZVAD (10 μM) or LIP-1 (200 nM). On day 15, cells were harvested for imaging under bright field (**l**), quantification of cell numbers (**m**) and released LDH (**n**). For **b**, **d**, **f**, **g**, **i**, **j**, **k**, **m** and **n**, each value represents mean ± s.e.m. of a triplicate. **o** On day 14 of differentiation, control and FASNᵒᵉ adipocytes were harvested and subjected to western blot using indicated antibodies. Statistical analysis was performed using two-sided unpaired Student's *t*-tests. Scale bar, 100 μm.

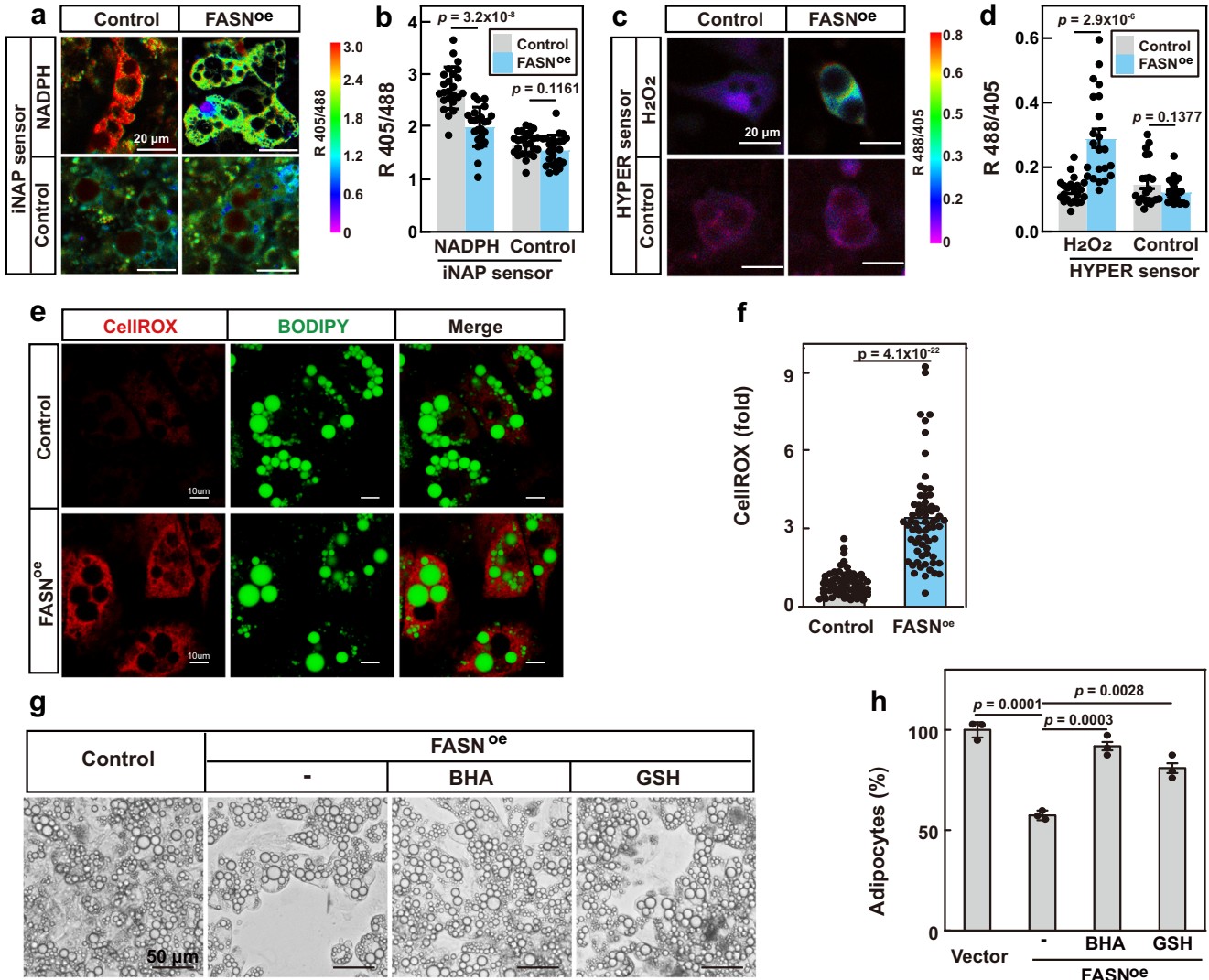

**Fig. 2 | Increased ROS production drives necroptosis of FASN-overexpressing adipocytes.** iNAP sensors for NADPH (**a**, **b**) or HYPER sensors for H₂O₂ (**c**, **d**) were introduced to control or FASNᵒᵉ 3T3-L1 cells by lentivirus. Cells were differentiated. On day 12, cells were imaged and data were processed as in *Methods*. **e**, **f** On day 12 of differentiation, control or FASNᵒᵉ 3T3-L1 adipocytes were treated with CellROX (2.5 μM) for 30 min and harvested for imaging as in *Methods*. **g**, **h** Starting from day

9 of differentiation, control and FASNᵒᵉ adipocytes were treated with BHA (50 μM) or GSH (1 mM). On day 15, cells were harvested for imaging under bright field (**g**) and quantification of the cell numbers (**h**). For **b**, **d**, each value represents mean ± s.e.m. from 20 cells. For **f**, each value represents mean ± s.e.m. from 70 cells. For h, each value represents mean ± s.e.m. from 3 samples. Scale bars are as indicated. Statistical analysis was performed using two-sided unpaired Student's *t*-tests.

knockdown of MED20 dramatically decreased the binding of RNA polymerase II on the promoters of *Snai1* and *Snai2* (Supplementary Fig. 2f). These results suggest that MED20 might be an upstream regulator of SNAIL and SLUG.

We went on to explore whether MED20 would control the transcription of *Fasn* through SNAIL and SLUG. We isolated primary

stromal vascular fractions (SVFs) from *Rosa-Cre^ERT2*;*Med20^f/f* mice, differentiated them into adipocytes, and induced deletion of *Med20* with 4-hydroxytamoxifen (4-OHT). As revealed by RNA sequencing (RNA-Seq), knockout of *Med20* decreased the transcription of *Snai1* and *Snai2*, but increased the transcription of *Fasn*, as well as *Acly* and *Acc1*, two other genes involved in de novo fatty acid synthesis (Fig. 3g). We

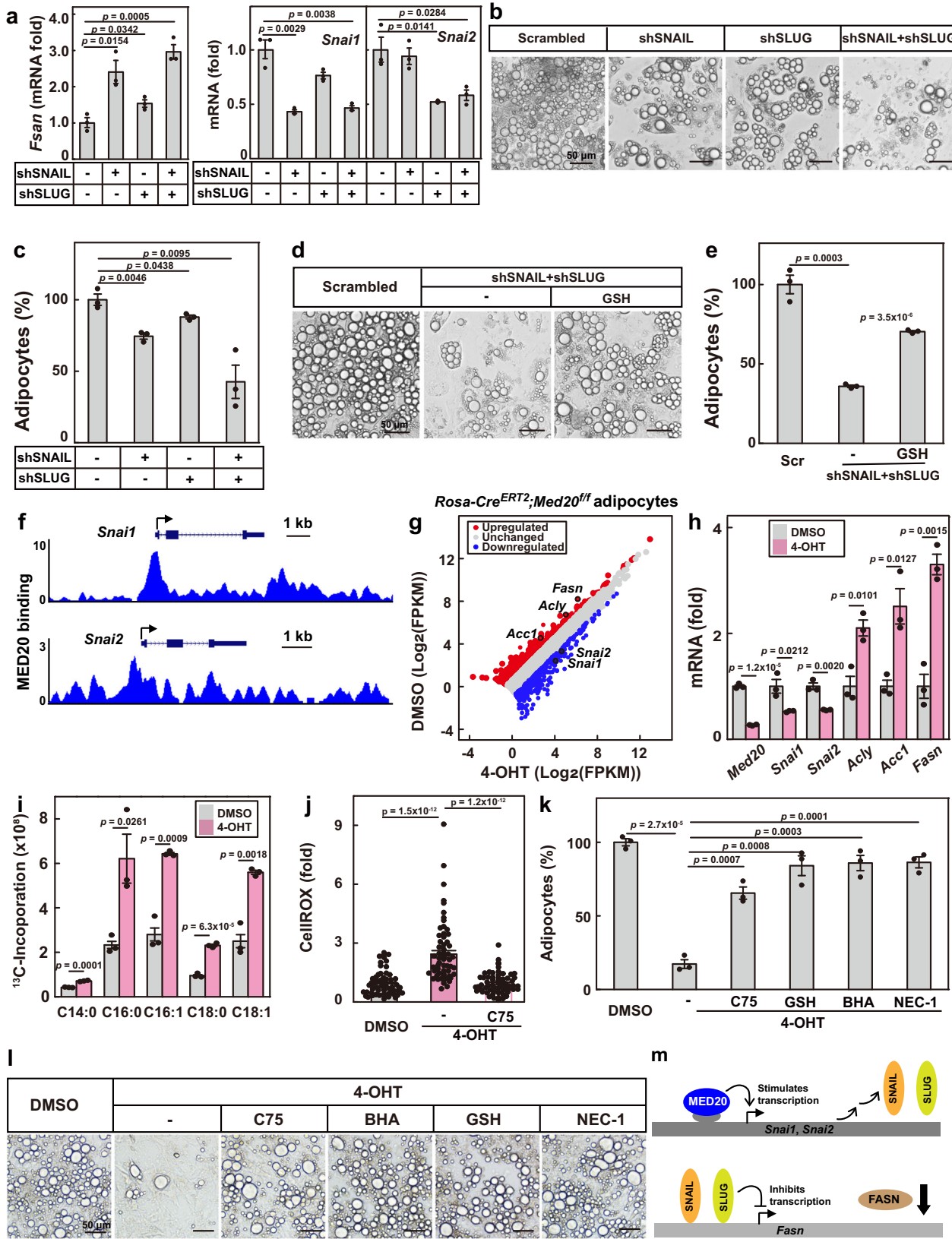

further validated the results by qRT-PCR and western blot analysis (Fig. 3h; Supplementary Fig. 2g, h).

We next examined the effect of MED20 on fatty acid synthesis and survival of the adipocytes. We performed $^{13}$C-glucose labeling and found that *Med20*-depleted adipocytes showed a 2- to 3-fold increase in the newly synthesized C16:0, C16:1, C18:0 and C18:1 (Fig. 3i;

Supplementary Fig. 2i, j). As indicated by CellROX and DCFDA, ROS production was also increased in *Med20*-depleted adipocytes, which was largely reversed by inhibiting FASN with C75 (Fig. 3j; Supplementary Fig. 2k). Similar to FASN overexpression or knockdown of SNAIL and SLUG, depletion of *Med20* caused cell death of the adipocytes, and it could be largely reversed by inhibiting FASN with C75, scavenging

**Fig. 3 | MED20 inhibits the transcription of *Fasn* through SNAIL and SLUG.**
**a**−**c** On day 4 of differentiation, 3T3-L1 cells were infected with lentivirus encoding the indicated shRNAs. **a** On day 9 of differentiation, cells were harvested for qRT-PCR analysis of the indicated genes. On day 14, cells were harvested for imaging under bright field (**b**) and quantification of cell numbers (**c**). **d**, **e** On day 9 of differentiation, control or SNAIL and SLUG double knockdown adipocytes were treated with GSH (1 mM) for 3 days. On day 14, cells were harvested for imaging under bright field (**d**) and quantification of cell numbers (**e**). **f** ChIP-Seq analysis of the binding of MED20 on the promoters of *Snai1* and *Snai2* in control and MED20 knockdown 3T3-L1 cells. The data were reanalyzed from GEO: GSE163281. **g**–**l** Primary SVFs were isolated from *Rosa-Cre^ERT2;Med20^f/f* mice, differentiated into mature adipocytes, and induced with 4-OHT to delete *Med20* on day 6. On day 9, cells were harvested and RNA was extracted for RNA-Seq (**g**) and qRT-PCR (**h**) analysis. **i** on day 9, de novo fatty acid synthesis was performed as in Fig. 1f. **j** On day 12, cells were treated with CellROX (2.5 μM) for 30 min and harvested for imaging as in Fig. 2e. **k**, **l** On day 9, cells were treated with C75 (50 μM), BHA (50 μM), GSH (1 mM) or NEC-1 (20 μM) for 3 days. On day 14, cells were harvested for quantification of cell numbers (**k**) and imaging under bright field (**l**). For a,c,e,h,I and k, each value represents mean ± s.e.m. from 3 samples. For **j**, Each value represents mean ± s.e.m. from 70 samples. **m** A summary of the findings in this figure. Scale bars are as indicated. Statistical analysis was performed using two-sided unpaired Student's *t*-tests.

ROS with BHA or GSH, or inhibiting necroptosis with NEC-1 (Fig. 3k, l). We also detected the marker proteins of necroptosis and found that there was a dramatic increase in p-MLKL (2.1-fold), p-RIPK1 (1.9-fold) and p-RIPK3 (3.9-fold) in *Med20*-depleted adipocytes (Supplementary Fig. 2l, m). The data further confirmed that knocking out *Med20* caused necroptosis of the adipocytes.

Taken together, MED20 is essential for the survival of adipocytes by inhibiting the transcription of *Fasn* via two transcriptional suppressors SNAIL and SLUG (Fig. 3m).

## Adipocyte-specific knockout of *Med20* triggers progressive lipodystrophy

We then sought to study the physiological significance of over-producing fatty acids in adipocytes. We generated adipocyte-specific *Med20* knockout (*Med20-AKO*) mice by crossing *Med20^f/f* mice with *AdipoQ-Cre* mice (Fig. 4a; Supplementary Fig. 3a). After 17 weeks on chow diet, *Med20-AKO* mice did not differ from control littermates in body weights (Fig. 4b); however, when subjected to glucose tolerance test and insulin tolerance test, *Med20-AKO* mice showed slower glucose clearance ability and were insulin resistant (Fig. 4c, d). Body composition analysis revealed that the fat mass in *Med20-AKO* mice (2.5 ± 0.1% of body weight) was significantly lower than that in control mice (7.6 ± 0.5% of body weight) (Fig. 4e). When dissected, the inguinal white adipose tissue (iWAT), gonadal WAT (gWAT) and brown adipose tissue (BAT) weighed significantly lighter and appeared smaller in *Med20-AKO* mice (Fig. 4f, g). In contrast, the liver of *Med20-AKO* mice weighed heavier and looked bigger than that in controls (Fig. 4f, g). Consistently, there was a 3-fold increase in liver triglyceride content in *Med20-AKO* mice (Fig. 4h; Supplementary Fig. 3b). Plasma levels of triglyceride and insulin were also significantly higher in *Med20-AKO* mice (Fig. 4h, i). As a reflection of reduced fat mass, plasma leptin level was reduced in *Med20-AKO* mice (Fig. 4i). These phenotypes mimicked the features of lipodystrophy.

We also took a close examination of the adipose tissues by H&E staining. In iWAT, gWAT and BAT of *Med20-AKO* mice, the number of adipocytes dramatically decreased, but the remaining adipocytes were much bigger than those in the control mice (Fig. 4j). Both mRNA and protein levels of de novo fatty acid synthesis genes were significantly higher in the iWAT of *Med20-AKO* mice (Supplementary Fig. 3d–f). There were also areas with clusters of nuclei in all the three types of adipose tissues of *Med20-AKO* mice (Fig. 4j), suggesting macrophage infiltration. To prove the point, we performed immunostaining using anti-Perilipin1 to indicate lipid droplet, and anti-Galectin3 to indicate macrophages. Indeed, large amounts of macrophages were observed in the adipose tissues of *Med20-AKO* mice (Fig. 4j). Consistently, the mRNA levels of the macrophage markers, F4/80, CD11c, IL-6 and IL-1β, were significantly upregulated in iWAT and gWAT of *Med20-AKO* mice (Supplementary Fig. 3c).

To further confirm that the adipocytes in *Med20-AKO* mice died of necroptosis, we performed immunohistochemistry using the p-MLKL antibody. As shown in Fig. 4k, the p-MLKL signal was much stronger in the iWAT and BAT of *Med20-AKO* mice compared with those of *Mde20^f/f* mice.

We have previously shown that knockout of *Med20* in the pre-adipocytes inhibits adipogenesis[27]. As defect in adipogenesis can cause lipodystrophy[6], one could argue that the *AdipoQ*-Cre might delete *Med20* in the preadipocytes, although it has been shown that the *AdipoQ*-Cre is highly specific to mature adipocytes and cannot induce gene deletion in preadipocytes[28,29]. To clarify the point, we isolated primary SVFs, and found that the protein level of MED20 in the pre-adipocytes of *Med20-AKO* mice did not differ from that of control mice (Supplementary Fig. 3g, h). However, after 9 days of differentiation, the protein level of MED20 was dramatically decreased in *Med20-AKO* adipocytes (Supplementary Fig. 3g, h). Thus, the *adipoQ-Cre* did not delete *Med20* in preadipocytes, but in adipocytes. Furthermore, on day 9 of differentiation, the Med20-AKO cells fully differentiated into mature adipocytes (Supplementary Fig. 3i–k), and the marker proteins of mature adipocytes, C/EBPα, PPARγ and Perilipin1, did not show any difference between control and *Med20-AKO* adipocytes (Supplementary Fig. 3g, h). On day 12 of differentiation, we noticed cell death of adipocytes from *Med20-AKO* mice. By day 15 and day 18, the majority of the *Med20*-deficient adipocytes died, while the control adipocytes survived well (Supplementary Fig. 3i–k). From these above, we conclude that the lipodystrophy of the *Med20-AKO* mice is not due to the deletion of *Med20* in preadipocytes.

To figure out when *Med20-AKO* mice started to develop lipodystrophy, we examined the mice at different ages. At 2 and 4 weeks old, the *Med20-AKO* mice did not show significant differences with their control littermates (Supplementary Fig. 4a–f). At 5 weeks old, the BAT of *Med20-AKO* mice started to weigh significantly less than that in control mice (Supplementary Fig. 4a). As revealed by H&E analysis, enlarged adipocytes and signs of inflammation started to show up in the BAT of *Med20-AKO* mice from 5 weeks old (Supplementary Fig. 4g). Starting from 7 weeks old, similar observations were made in the iWAT and gWAT of *Med20-AKO* mice (Supplementary Fig. 4b, c; h, i). Consistent with the decreased mass of WAT, plasma leptin levels in *Med20-AKO* mice started to be significantly lower from 7 weeks old (Supplementary Fig. 4d). Starting from 9 weeks old, the liver of *Med20-AKO* mice weighed heavier, and contained more triglycerides (Supplementary Fig. 4e, f and j), indicating ectopic lipid storage.

Taken together, adipocyte-specific knockout of *Med20* induces progressive lipodystrophy in mice.

## Scavenging ROS or inhibiting necroptosis largely reverses lipodystrophy in *Med20-AKO* mice

To further consolidate the results, we next sought to examine whether we could rescue the lipodystrophic phenotype by scavenging ROS or inhibiting necroptosis. We first treated 6-week-old *Med20-AKO* mice with a daily gavage of BHA for 8 weeks (Fig. 5a). Figure 5b shows that BHA treatment did not have much effect on body weights (Fig. 5b); however, it significantly improved glucose clearance rate and insulin sensitivity of *Med20-AKO* mice (Fig. 5c, d). Body composition analysis revealed that BHA treatment significantly increased the fat mass of

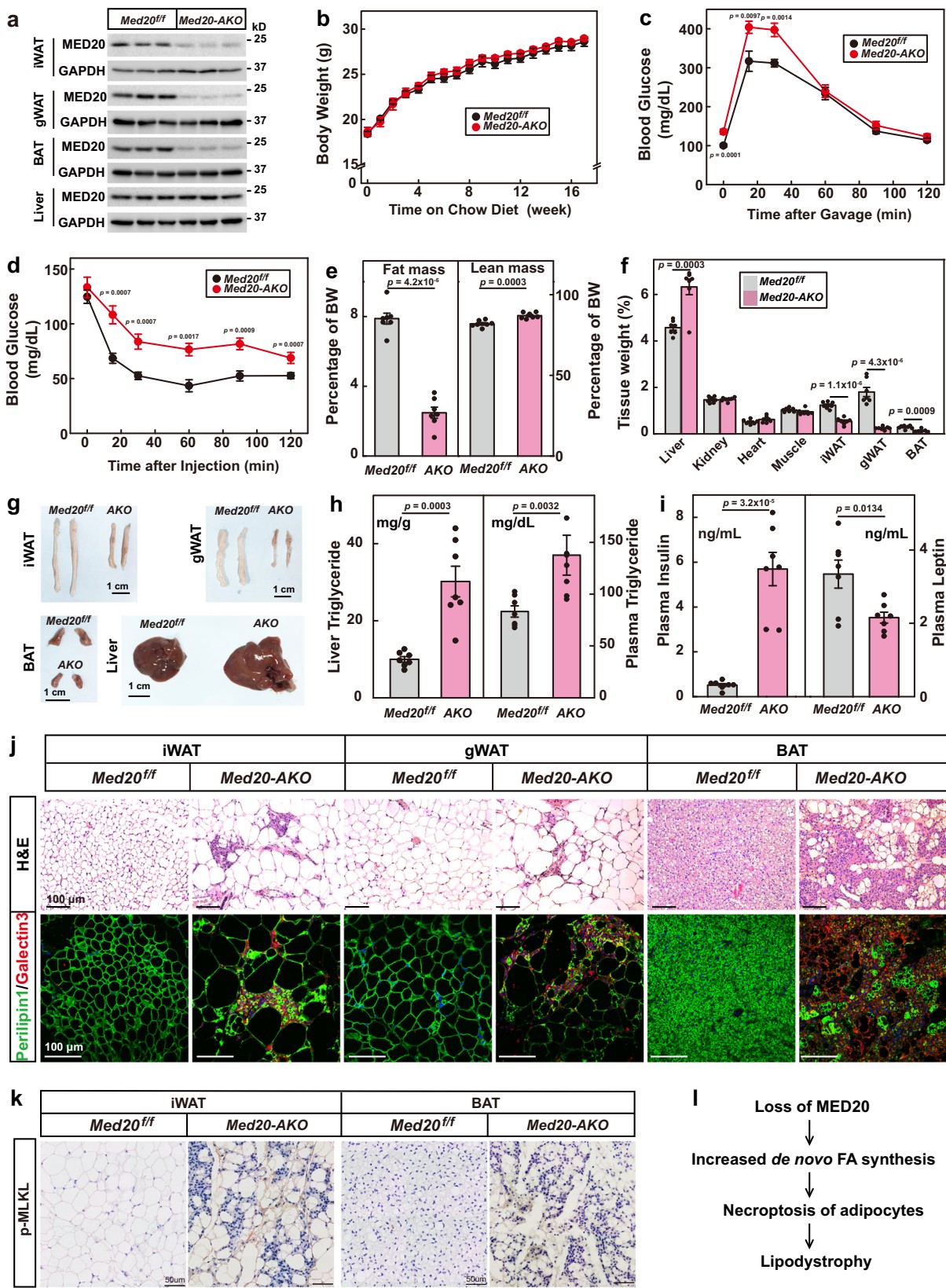

*Med20-AKO* mice (Fig. 5e), which was further illustrated by the increased sizes of iWAT and gWAT (Fig. 5f). Furthermore, the elevated levels of plasma and liver triglyceride in *Med20-AKO* mice were also largely normalized by BHA treatment (Fig. 5g). We also noticed an increase in plasma $H_2O_2$ level in *Med20-AKO* mice, which was brought down to normal by BHA treatment (Fig. 5h). H&E and

immunofluorescence analysis revealed that BHA treatment dramatically reduced macrophage infiltration in iWAT (Fig. 5i). Overall, scavenging ROS largely reverses the lipodystrophy in *Med20-AKO* mice (Fig. 5j).

We then explored whether we could rescue the lipodystrophy in *Med20-AKO* mice by inhibiting necroptosis with NEC-1 (Supplementary

**Fig. 4 | Adipocyte-specific knockout of *Med20* triggers progressive lipody-strophy. a–k** *Med20[f/f]* mice were crossed with *AdipoQ*-Cre mice to generate adipose tissue-specific *Med20* knockout mice (*Med20-AKO*). **a** Proteins were extracted from iWAT, gWAT, BAT and liver of 3 pairs of *Med20[f/f]* and *Med20-AKO* mice, and subjected to western blot analysis using anti-MED20 and anti-GAPDH antibodies. **b–j** Male *Med20[f/f]* and *Med20-AKO* mice (4-week-old, *n* = 7) were subjected to chow diet for 17 weeks. **b** Body weight was monitored every week. Glucose (**c**) and insulin (**d**) tolerance tests were performed on week 12 and week 14, respectively. Body composition (**e**) was analyzed on week 17. After that, mice were euthanized, and tissues were collected and weighed (**f**). Represent images of iWAT, gWAT, BAT and liver were shown (**g**). Liver triglyceride (**h**, left), plasma triglyceride (**h**, right), insulin (**i**, left) and leptin (**i**, right) were measured. **j** iWAT, gWAT and BAT were fixed, sliced and subjected to H&E (upper panel) and immunofluorescence using anti-Perilipin and anti-Galectin3 antibodies (lower panel). **k** Immunohistochemistry was performed in iWAT (7 weeks old) and BAT (5 weeks old) in control and *Med20-AKO* mice using anti-Phospho-MLKL antibody. **l** A schematic summary of the key findings in this figure. For **b–f**, **h** and **i**, each value represents mean ± s.e.m. from 7 samples. Scale bars are as indicated. Statistical analysis was performed using two-sided unpaired Student's *t*-tests.

Fig. 5a). Similar to BHA, NEC-1 treatment significantly improved the metabolic disorders in *Med20-AKO* mice and largely reversed the lipodystrophy (Supplementary Fig. 5b–i).

These studies further indicate that scavenging ROS or inhibiting necroptosis can rescue the lipodystrophy in *Med20-AKO* mice.

### Scavenging ROS reverses stavudine-induced partial lipodystrophy

We then sought to investigate whether the mechanism we identified could be applied to other forms of acquired lipodystrophies. HIV-infected patients with highly active antiretroviral therapy tend to develop partial lipodystrophy, characterized by selective loss of subcutaneous fat[9]. Nucleoside analog reverse transcriptase inhibitors (NRTIs) are responsible for the most severe form of HIV-associated lipodystrophies[30]. NRTIs are shown to cause mitochondrial dysfunction and increase oxidative stress in adipocytes[31]. Patients with HIV-associated lipodystrophy have higher levels of lipid and protein oxidation in the plasma[15]. However, it remains unclear whether increased ROS production is a cause or consequence of HIV-associated lipodystrophy.

To address the issue, we treated adipocytes with stavudine, an NRTI that is still used as a first-line treatment under settings of limited resource. We found that stavudine treatment caused cell death of adipocytes and increased ROS production (Fig. 6a–c; Supplementary Fig. 6a, b). Furthermore, GSH treatment largely rescued stavudine-induced cell death (Fig. 6a–c; Supplementary Fig. 6a, b). We noticed that stavudine did not have much effect on the expression of de novo fatty acid synthesis genes (Supplementary Fig. 6c), suggesting that stavudine might increase ROS production through other mechanisms. We also confirmed that stavudine-induced cell death is necroptosis, as NEC-1, but not ZVAD or LIP-A, rescued the cell death (Supplementary Fig. 6d–f).

To further validate the above results, we generated a mouse model to mimic HIV-associated lipodystrophy by treating HFD-fed mice with stavudine for 9 weeks (Fig. 6d). Although having no effect on body weights (Supplementary Fig. 6g), stavudine treatment significantly impaired glucose clearance and insulin sensitivity of the lipodystrophy mice, which was largely reversed by daily treatment with GSH (Fig. 6e, f). At the end of the experiment, stavudine-treated mice showed selective loss of iWAT (Fig. 6g, h; Supplementary Fig. 6h), similar to that in patients with HIV-associated partial lipodystrophy[30]. GSH treatment largely restored the fat mass of iWAT (Fig. 6g, h). A further examination by H&E and immunostaining revealed that stavudine-treated mice showed increased adipocyte size and macrophage infiltration in iWAT, which could be largely reversed by GSH treatment (Fig. 6i). Stavudine also caused an increase in liver triglyceride content and plasma $H_2O_2$ level, which was also normalized by GSH treatment (Fig. 6j, k; Supplementary Fig. 6i).

These results indicate that increased ROS production in adipocytes is the major cause of lipodystrophy in stavudine treated HIV patients, and that scavenging ROS with glutathione might be a potential therapeutic strategy to reverse the side effect of the NRTIs in HIV-infected patients (Fig. 6l).

### Glutathione treatment improves metabolic disorders in a patient with acquired lipodystrophy

We moved on to examine whether increased ROS production might also drive the onset of other forms of acquired lipodystrophy. We admitted a 48-year-old female patient with lipodystrophy to the hospital in August, 2020. She had a normal BMI (20.5 kg/m[2]), but looked very skinny and muscular (Fig. 7a). Body composition analysis showed that her fat mass index (FMI) was 2.43 kg/m[2], way below the normal range (5-9 kg/m[2]). Her fasting blood glucose was 6.31 mM, and HOMA-IR was 2.09. She also had hypertriglyceridemia (2.76 mM). From a fribroscan analysis (CAP, 305) and liver biopsy, she was diagnosed with nonalcoholic steatohepatitis (NASH) (Supplementary Fig. 7a, b). According to the patient, she started to notice loss of subcutaneous fat in 2015. The patient did not have a family history of metabolic disorders. We performed whole-genome exon sequencing of the patient, her brother and parents, but did not find any novel mutations that might lead to lipodystrophy (Supplementary Fig. 7c). From all these above, the patient was diagnosed with acquired generalized lipodystrophy.

The patient was on different medications from August 2020 to November 2021. Her plasma triglyceride level was well controlled, decreasing from 2.76 to 1.76 mM. Her CAP score dropped to 263 dB/m (Supplementary Table 2), but it was still at a high level, steatosis grade 2, reflecting that 34-66% of her hepatocytes contain fat vacuoles. Her ALT and AST values were 70.4 U/L and 43 U/L, respectively (Supplementary Table 2), indicating liver damage.

Of note, the patient runs a hair salon where she is constantly exposed to oxidants in hair dye materials. We therefore reasoned that increased oxidative stress might be the potential cause of her acquired lipodystrophy. Indeed, her plasma $H_2O_2$ level was higher than normal (Fig. 7b). Based on the liver dysfunction, which was proven by liver biopsy and biochemical tests, we further added GSH (0.4 g, tid) on top of her previous prescription and evaluated her conditions 6 months later. Figure 7b shows that her plasma $H_2O_2$ level fell to the normal range after GSH treatment. Her body weight increased from 52.4 to 53.25 kg (Supplementary Table 2). Her total body fat and fat mass index also increased (Fig. 7c), which was further confirmed by the increase in the subcutaneous fat (Fig. 7d). Fibroscan analysis revealed that her CAP score decreased from 263 to 240 (Fig. 7e), indicating an improvement in her fatty liver condition. MRI-PDFF analysis further confirmed that 6 of the 9 regions of her liver showed decreased fat fraction after GSH treatment (Fig. 7f, g). Furthermore, her plasma levels of ALT, AST and HbA1c dropped to normal ranges after GSH treatment (Fig. 7h, i). These results indicate that GSH treatment improves the metabolic disorders of the patient (Fig. 7j). Therefore, scavenging ROS might be a potential strategy to manage acquired generalized lipodystrophy.

## Discussion

Despite that adipose tissue is the primary site for fatty acid storage, the majority of the stored fatty acids is not made by itself[2,3]. The physiological significance of such a phenomenon remains unclear. Here, we show that elevated de novo fatty acid synthesis increases ROS production and causes necroptosis in adipocytes. We first activated the transcription of FASN with the CRISPR-based activation system.

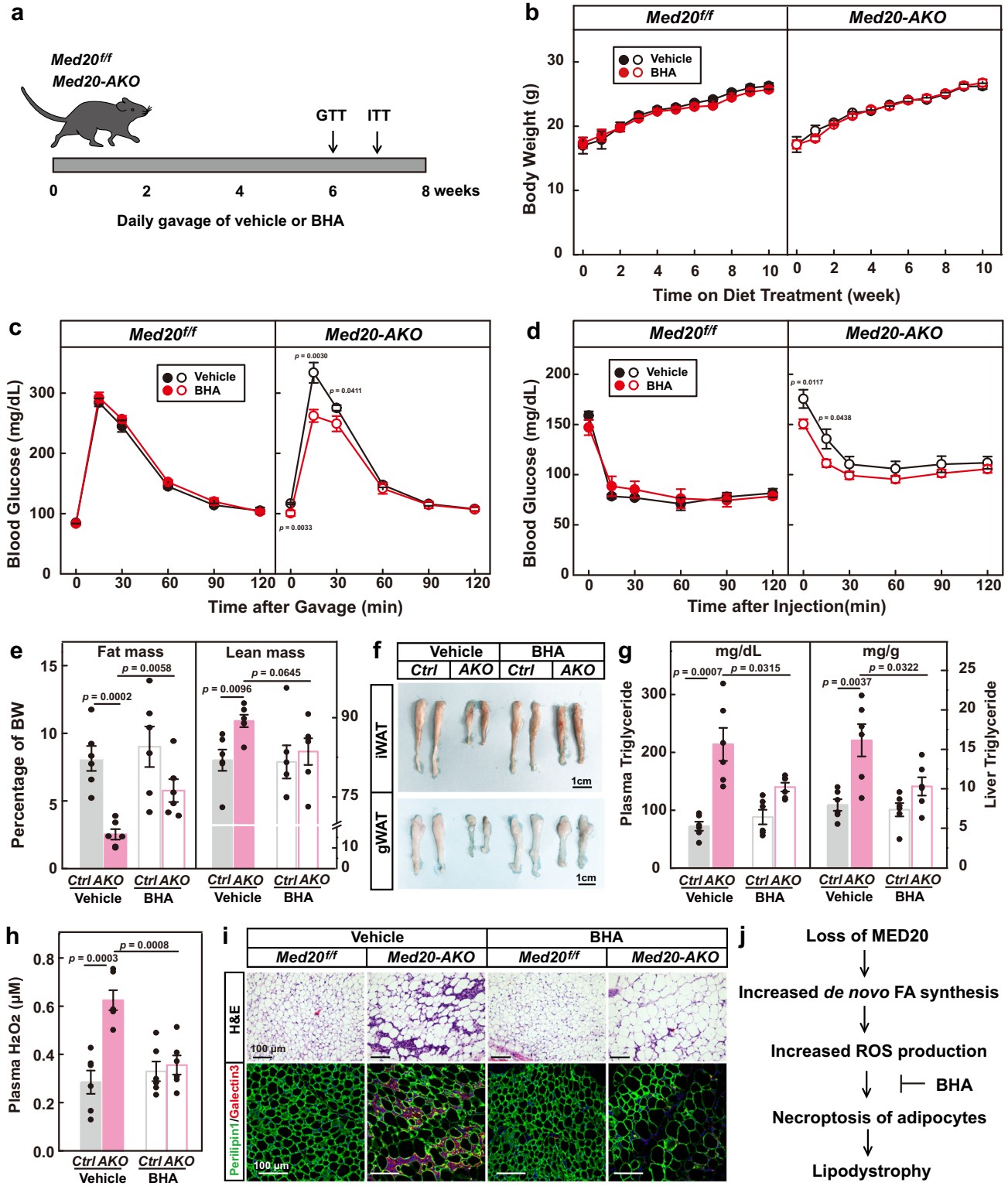

**Fig. 5 | Scavenging ROS reverses lipodystrophy in *Med20-AKO* mice. a** Chow-fed male *Med20^{f/f}* and *Med20-AKO* mice (6-week-old, $n = 6$) were daily gavaged with vehicle or BHA (10 mg/ml in corn oil) for 8 weeks. **b** Body weights were monitored every week for 8 weeks. Glucose (**c**) and insulin (**d**) tolerance tests were performed on week 7 and 8, respectively. **e** Body composition was analyzed on week 8. **f** On week 8, mice were euthanized. Representative images of iWAT and gWAT were shown. **g** Plasma and liver triglyceride levels were measured. **h** Plasma $H_2O_2$ level was measured. **i** iWAT was subjected to H&E and immunostaining using anti-Perilipin and Galectin3 antibodies as in Fig. 4j. Each value represents mean ± s.e.m. of 6 mice. **j** A schematic summary of the key findings in this figure. Scale bars are as indicated. Statistical analysis was performed using two-sided unpaired Student's *t*-tests.

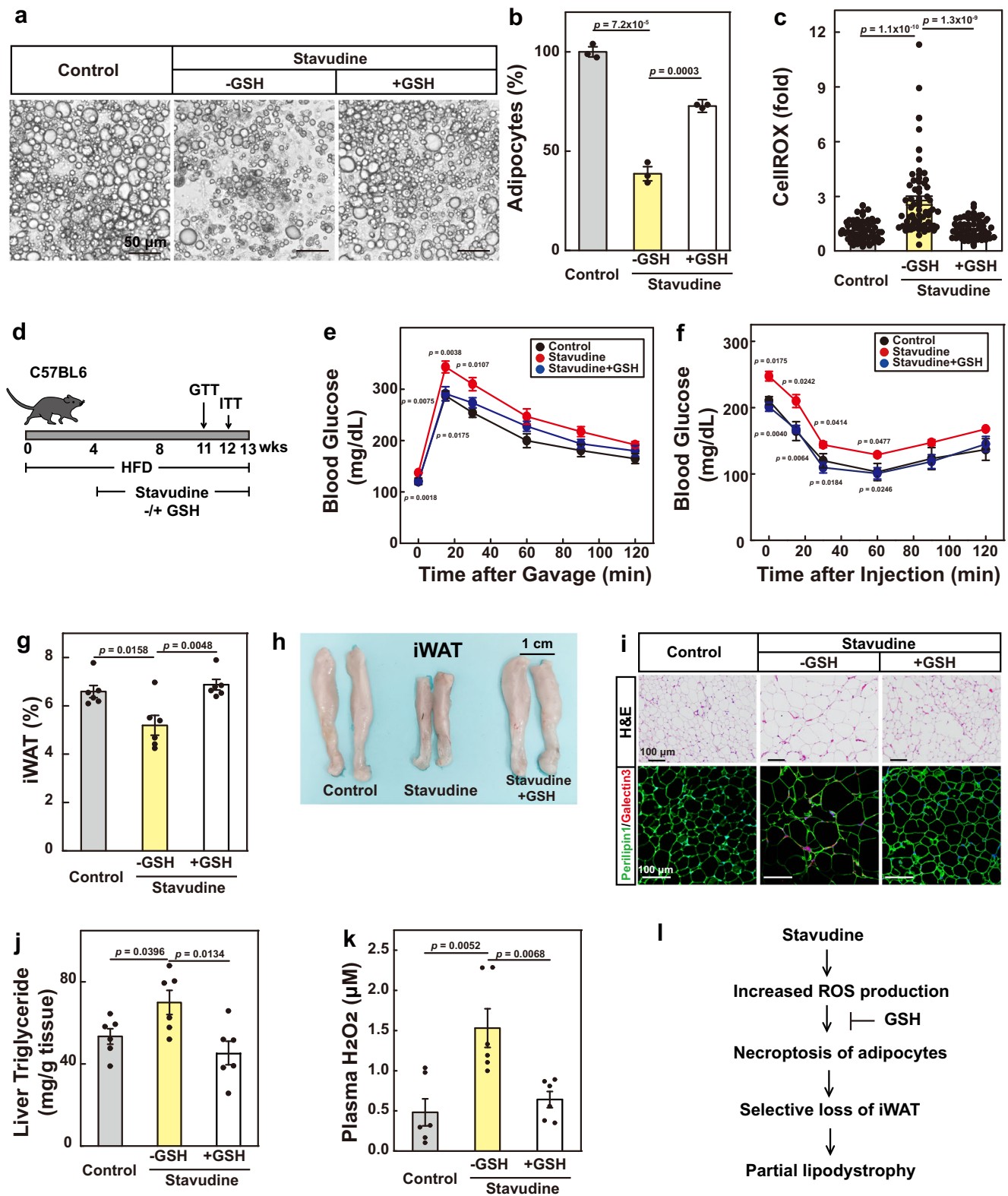

Although the FASN-overexpressing cells synthesized more fatty acids, they showed decreased cellular content of NADPH and subsequently increased ROS production, which ultimately led to necroptosis of the adipocytes. We then identified MED20 as a negative regulator of endogenous transcription of FASN. Loss of MED20 in adipocytes significantly upregulated the mRNA levels of FASN and increased de novo fatty acid synthesis. MED20-deficient adipocytes died of increased ROS production, and *Med20-AKO* mice developed lipodystrophy. Consistent with our results, adipocyte-specific overexpression of SREBP-1c, a master regulator of lipid synthesis genes including *Acly*, *Acc1* and *Fasn*, causes congenital lipodystrophy[32]. SREBP1c has also been suggested to play a key role in the pathogenesis of human lipodystrophies[33]. In addition, increased FASN expression in human adipose tissues is linked to insulin resistance and inflammation[34].

**Fig. 6 | Glutathione reverses stavudine-induced partial lipodystrophy in mice.** a–c Primary SVFs from WT mice were differentiated into mature adipocytes. On day 0, cells were treated with stavudine (1 mM) with or without GSH (1 mM). On day 6, cells were harvested for imaging under bright field (**a**) and quantification of cell numbers (**b**). For **b**, each value represents mean ± s.e.m. of 3 samples. On day 4, cells were treated with CellROX (2.5 μM) for 30 min and harvested for ROS quantification (**c**). For **c**, each value represents mean ± s.e.m. of 70 samples. d–k C57BL6 mice were subjected to HFD for 4-weeks, and then treated with stavudine (5 mg/mL in water) with daily gavage of GSH (100 mg/kg) for 9 more weeks. Glucose (**e**) and insulin (**f**) tolerance tests were performed on week 11 and 12, respectively. On week 13, mice were euthanized. Percentage of iWAT/body weight was plotted in (**g**). Representative images were shown in (**h**). **i** H&E and immunostaining of iWAT using anti-Perilipin and Galectin3 antibodies were performed as in Fig. 4j. **j** Liver trigly-ceride levels were measured. **k** Plasma H$_2$O$_2$ levels were measured. **l** A schematic summary of the key findings of the figure. Each value represents mean ± s.e.m. of 6 mice. Statistical analysis was performed using two-sided unpaired Student's t-tests.

Therefore, it is reasonable to believe that the low de novo fatty acid synthesis rate in adipocytes is a protective strategy to avoid oxidative stress-induced cell death.

It should be noted that liver is the primary site for de novo fatty acid synthesis; however, the hepatocytes, unlike the adipocytes, are able to manage the ROS induced by de novo fatty acid synthesis. A possible explanation is that hepatocytes have the highest GSH content in the body, which can go up to 10 mM, whereas other cell types have a GSH concentration around 1-2 mM[35–37]. The high GSH content might help to manage the oxidative stress caused by high rates of de novo fatty acid synthesis.

We have also identified increased ROS production as a potential cause of acquired lipodystrophy. As lipodystrophy is a rare disease, little is known about the underlying mechanisms and the management strategies remains poor. Here, we found that *Med20-AKO* mice progressively developed lipodystrophy, which mimic the features of acquired generalized lipodystrophy and can be rescued by ROS scavenging. We generated another acquired lipodystrophy mouse model using stavudine and demonstrate that stavudine induces partial lipodystrophy through increasing ROS production. Importantly, in a patient with acquired lipodystrophy, we showed that supplementation of glutathione facilitated management of metabolic disorders.

In agreement with our findings, in a clinical study in HIV-infected patients receiving antiretroviral therapy, 24-week treatment of vitamin E, a lipophilic antioxidant, significantly decreased levels of plasma alanine aminotransferase and cytokeratin 18, and CAP scores, indicating improved NASH[38]. It should be noted that in obese people with NASH, vitamin E has been shown to be an effective treatment for patients without type 2 diabetes[39], but not for those with type 2 diabetes[40]. As many lipodystrophic patients also develop type 2 diabetes, more clinical studies might be required to evaluate the therapeutic potential of vitamin E in treating HIV-associated lipodystrophies.

Notably, due to the rarity of acquired lipodystrophy, we only recruited one patient in the study. Thus, there might be some uncertainty to apply the observation in the current study to other patients with acquired lipodystrophy. However, based on the evidences we obtained from mouse studies and the metabolic benefits of antioxidant in HIV-infected patients receiving antiretroviral therapy[38], it is reasonable to deduce that increased oxidative stress might be a potential cause of acquired lipodystrophies. It is also promising that GSH might be a therapeutic treatment for patients with acquired lipodystrophy.

It has been well documented that long-term engagement in occupations involving regular exposure to oxidants would cause an increase in plasma ROS levels[41]. However, in the population-based epidemiological studies, BMI and other common biological metabolic evaluations are used, without providing a precise evaluation of body fat distribution; therefore, the association has not been established between long-term engagement in occupations involving regular exposure to oxidants and lipodystrophy. In the future, it will be interesting to investigate the causal-effect study about oxidants exposure and fat mass and adipocyte function in humans.

Our studies have also uncovered a new role of MED20 in regulating the physiological function of adipose tissues. Although MED20 is a common component of the Mediator complex, it has been shown to be a non-essential subunit of the Mediator complex[42,43]. We have previously shown that MED20 act as a functional bridge between C/EBPβ and RNA polymerase II to control the transcription of C/EBPα and PPARγ, thereby regulating adipogenesis[27]. Here, we show that MED20 controls the transcription of *Snai1* and *Snai2* to inhibit the transcription of *Fasn*. Taken together, we speculate that in different contexts, MED20 might interact with different transcription factors to control the transcription of downstream genes. Notably, a previous study reported a homozygous mutation in MED20 in human[44]. The mutation-bearing siblings were diagnosed with infantile-onset spasticity and childhood-onset dystonia, progressive basal ganglia degeneration and the brain atrophy[44]. It was not documented about whether the patients developed lipodystrophy, which would be interesting to know in the future.

Macrophages play a pivotal role in the homeostasis of adipose tissues[45]. In both WAT and BAT, we observed large amounts of macrophage infiltration and elevation of inflammatory markers in *Med20-AKO* mice. The inflammation was dramatically improved after treating the mice with ROS scavenging reagent or necroptosis inhibitors. Therefore, we speculate that the macrophages are recruited by the dead adipocytes. In addition to clean the dead adipocytes, the macrophages might also help to regenerate the adipose tissues[46].

Interestingly, preventing necroptosis using NEC-1 also improves the lipodystrophy phenotype and comorbidities; however, inhibition of necroptosis will presumably not resolve the cellular ROS stress induced by MED20 deficiency. We noticed that the surviving adipocytes were much bigger in size and the inflammation markers were decreased in NEC-1-treated adipose tissues. Therefore, inhibition of necroptosis likely does not cause other types of cell death of the adipocytes, at least in the 8-week period of the experiment. Currently, we do not have a clear answer about how these adipocytes live with the ROS stress. Presumably, the adipocytes might develop a compensatory mechanism to resolve the ROS stress, or they might eventually die when the ROS levels accumulates to a certain level.

In summary, our findings uncover the physiological significance of the low fatty acid synthesis rate in adipocytes, identify a potential cause of acquired lipodystrophy and may provide an effective means for lipodystrophy management.

## Methods

We obtained dexamethmasone, isobutylmethylxanthine (IBMX), pioglitazone, bovine insulin, 4-hydroxy tamoxifen (4-OHT), SDS, DTT, DMSO and Triton X-100 from Sigma-Aldrich; Dulbecco's modified Eagle's medium (DMEM) with low (1 g/L) or high (4.5 g/L) glucose, fetal and neonatal bovine serum, and puromycin from Thermo Fisher Scientific; protease inhibitor cocktail from Roche Applied Science; and all other chemicals from local suppliers unless otherwise specified. For knockdown of genes, a pLKO.1 vector (Addgene, 10878)[47] was used. The primer sequences are listed in Supplementary Table 1.

### Patient information

The patient was diagnosed with acquired lipodystrophy at the first affiliated hospital of Nanjing Medical University in August, 2020. She was on pioglitazone (30 mg, qd) from August to October in 2020,

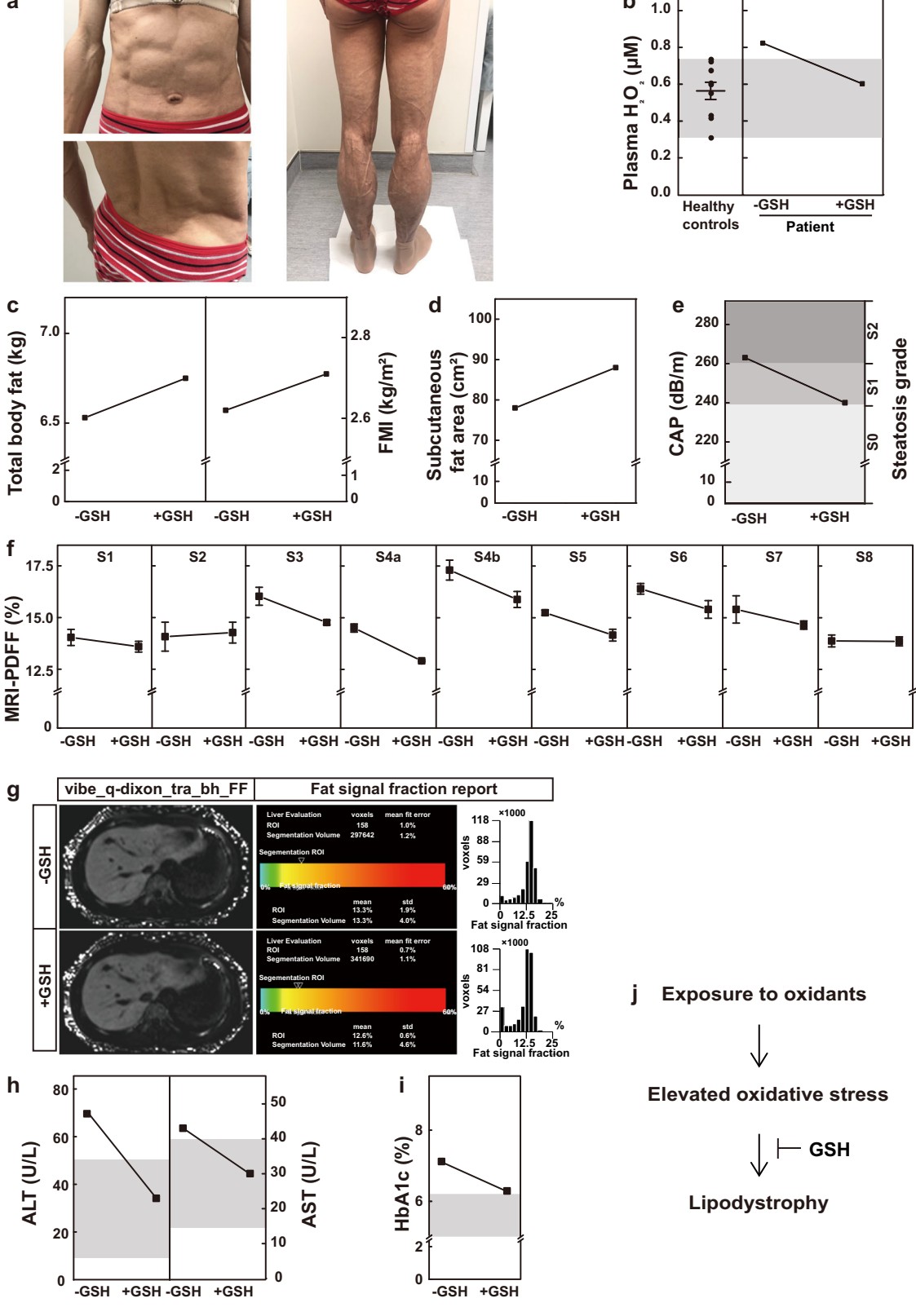

**Fig. 7 | Glutathione improves the metabolic disorders in a patient with acquired lipodystrophy. a** The pictures of a female patient who was diagnosed of acquired general lipodystrophy. **b** Her plasma $H_2O_2$ levels were measured before and after GSH treatment. For healthy control, the value represent mean ± s.e.m. of 9 samples. **c** Her total body fat was analyzed by DXA (**c**, left), and fat mass index (FMI) was calculated (**c**, right). **d** Subcutaneous fat area (cm²) was measured by a fat monitor. **e** CAP was analyzed by fibroscan. **f**, **g** Liver fat content was analyzed by MRI-PDFF. Each value represents mean ± s.e.m. of a triplicate. Plasma ALT (**h**, left), AST (**h**, right) and HbA1c (**i**) were measured. **j** A summary of the key findings in the figure.

gliclazide (60 mg, qd) from October in 2020 to April in 2021, and switched to LANTUS insulin glargine injection (20 u, qn) from April to November in 2021. From November in 2021, on top of LANTUS insulin glargine injection (20 u, qn), she was on GSH (0.4 g, tid). GSH is a standard and widely used choice for treating liver dysfunction. The CAP and LSM scores were measured by fibroscan (Echosens Fibroscan PRO). The visceral and subcutaneous fat areas were measured by OMRON HDS-2000. Total body composition was analyzed by DXA (Hologic DXA Discovery). Fat content in different regions of the liver was analyzed by MRI-PDFF (Siemens magnetom vida). The observational study protocol was reviewed and approved by the Human Ethics Committee of the First Affiliated Hospital of Nanjing Medical University (2019-SR-122). Written informed consent was obtained from the patient.

### Mice
All mice were housed in colony cages at 22℃ with 12-h light/12-h dark cycles. The dark cycle began at 7 pm. All animal studies were performed with the approval of the Institutional Animal Care and Research Advisory Committee at Fudan University and Xiamen University. Mice were euthanized by isoflurane at the appropriate time during the study and tissue sample were removed for further experiments.

*Med20*[f/f] mice were previously generated by our laboratory[27], and they were bred with an *AdipoQ-Cre*[28] mice to generate adipocyte-specific *Med20* knockout mice (*Med20-AKO*). *Med20*[f/f] mice were also crossed with *Gt(ROSA)26Sor*[tm1(cre/ERT2)Tyj] mice[48] to generate inducible *Med20* knockout mice, which was used for isolation of primary stromal vascular fractions.

The chow diet (Xietong Organism, Nanjing, China) and high-fat diet (HFD, Research Diet, D12492) contain 12% and 60% of calories from fat, respectively.

### Glucose and insulin tolerance tests
Glucose and insulin tolerance tests were performed as previously described[49,50]. Briefly, for oral glucose tolerance test, mice were fasted from 5 pm to 9 am and gavaged with D-glucose at 2 mg/kg (chow) or 1 mg/kg (HFD). For insulin tolerance test, mice were fasted from 8 am to 2 pm and intraperitoneally injected with insulin at 0.5 U/kg (chow) or 1 U/kg (HFD). Blood was collected from tail vein at 0, 15, 30, 60, 90 and 120 min after gavage of glucose or injection of insulin.

### Histology, immunofluorescent and immunohistochemistry analysis
At the end of the experiments, adipose tissues and liver were collected and fixed in 4% (wt/vol) paraformaldehyde in PBS for 20-48 h. The fixed tissues were embedded in paraffin and sectioned at 5 µm. Hematoxylin and eosin (H&E) staining was performed as previously described[51].

Immunofluorescence was performed as previously described[27]. Briefly, slides were deparaffinized by boiling in 10 mM sodium citrate, blocked with 10% goat serum, 0.1% Triton X-100 and 1% BSA in PBS for 30 min at 37℃, and incubated with antibodies of Perilipin1 (1:200, CST, #9349 s) and Galectin3 (1:200, Protein Tech, #60207-1-Ig). The slides were visualized on a Zeiss LSM-880+ confocal microscope.

Immunohistochemistry of p-MLKL was performed as previously described[20]. Briefly, slices were deparaffinized and boiled in 10 mM sodium citrate to unmask the antigens following a standard protocol. Slices were blocked with 0.1% Triton X-100 and 3% BSA in PBS for 30 min at room temperature, followed by incubation with anti- p-MLKL (1:250; Abcam, ab187091) overnight at 4℃. On the second day, slides were incubated with diluted HRP-conjugated secondary antibody for 50 min at room temperature, followed by adding DAB substrate (Beyotime, #P0202) for 3 min as manufacturer's guide. The slides were visualized on an Evident Olympus VS200 microscope.

### Measurements of metabolic parameters
Body composition was monitored on an Echo-MRI analyzer. Plasma levels of insulin, leptin, and $H_2O_2$ were measured using commercial Elisa kits from EZassay (#MS100), Cayman Chemical (#A05176), and Solarbio (#BC3595), respectively. Liver triglyceride was extracted using the Folch method. Triglyceride was measured using a commercial kit from Wako (#290-63701).

### Isolation of primary stromal vascular fractions
Inguinal white adipose tissue (iWAT) was dissected from 4-week old male *Rosa-Cre*[ERT2];*Med20*[f/f] mice and subjected stromal vascular fractions (SVFs) isolation as previously described[51,52]. Briefly, inguinal WAT was minced and digested with 1 mg/ml collagenase II (C6885, Sigma-Aldrich) at 37℃ for 1 h. The digested cells were neutralized with Medium A (Dulbecco's modified Eagle's medium (DMEM) with low glucose (1 g/L)) and filtered with a 100-µm nylon cell strainer (BD Biosciences). Cells were then resuspended in Medium A, filtered with a 40-µm nylon cell strainer (BD Biosciences) and plated into 6-well plates in Medium A at 37℃ in an atmosphere of 8.8% $CO_2$. SVFs were immortalized with lentivirus encoding SV40 large T antigen as previously described[53].

### Culture and differentiation of preadipocytes
3T3-L1 preadipocytes and SVFs were cultured in Medium A at 37℃ in an atmosphere of 8.8% $CO_2$ and maintained in less than 50% confluence. Cells were subjected to a standard cocktail hormone as previously described[49,51]. After day 8, mature adipocytes were cultured in Medium B (Dulbecco's modified Eagle's medium (DMEM) with high glucose (4.5 g/L), 10% fetal bovine serum, 100 U/ml penicillin, and 100 mg/ml streptomycin), and fresh medium was changed every two days. For inducible deletion of *Med20*, SVFs-derived adipocytes (*Rosa-Cre*[ERT2];*Med20*[f/f]), on day 6 of differentiation, were treated with 4-OHT (10 µM) for 4 days. On day 14, cells were harvested for imaging under bright field. The cultured medium was also collected for measurement of released LDH (Beyotime, #C0016).

### Lentivirus production and infection
To knock down genes in 3T3-L1 adipocytes, indicated shRNAs were cloned into pLKO.1 vector (Addgene, 10878), and lentivirus was produced in HEK293T cells as previously described[50,51]. Briefly, HEK293T cells were co-transfected with the related plasmid with psPAX2 (Addgene, 12260) and pMD2.G (Addgene, 12259). Lentivirus-containing medium were collected, aliquoted and stored at −80℃ until use. Lentivirus expressing iNap sensors of NADPH and its control mutant were as previously described[21]. For expression of iNap sensors, preadipocytes were infected, and later differentiated into mature adipocytes for further study. For knockdown of genes, cell on day 4 of differentiation were infected, and further differentiated to mature adipocytes for indicated studies.

### CRISPR-based activation
The CRISPR-based activation system was adopted from a published protocol[18]. To overexpress *Fasn* in 3T3-L1 adipocytes, the indicated sgRNA was cloned into lentiSAMv2 (Addgene, 75112). Lentiviruses including lentiSAMv2 and lentiMPHv2 (Addgene, 89308) were produced in HEK293T cells as mentioned above. Preadipocytes were infected and selected with Hygromycin B and Blasticidin for 6 days, and differentiated into mature adipocytes for further study.

### Visualization and quantification of intracellular reactive oxygen species
Adipocytes after 12 days of differentiation were used for the analysis, when both control and MED20-deficient adipocytes were alive. To

visualize intracellular reactive oxygen species (ROS), cells were treated with CellROX (2.5 µm, Thermo fisher, #C10422) and BODIPY 493/503 (10 µm, Thermo Fisher, #D3922) for 30 min at 37 °C. After wishing 3 times with PBS, cells were harvested and imagined on Zeiss LSM-880+ confocal microscope. DCFDA is also used to measure cellular ROS. Cell were treated with DCFDA (25 µM, Abcam, #ab113851) for 45 min at 37 °C. To quantify the DCFDA signal, cells were treated with DCFDA as above. Cells were then harvested and the fluorescent intensity of DCFDA was quantified on a Tecan Spark plate reader at Ex/Em = 485/535 nm.

### Visualization and quantification of intracellular NADPH and $H_2O_2$

A genetically encoded fluorescent indicator of NADPH (iNAP1) or its mutant that lost NADPH binding (iNAPc)[21], or HYPER sensors to indicate $H_2O_2$[23], was introduced to immortalized preadipocytes (*Rosa-Cre^ERT2;Med20^{f/f}*) by lentivirus. After 8 days of differentiation, *Med20* was inducibly deleted by 4-OHT (8 µM). On day 12, cells were harvested and imaged on Zeiss LSM-880+ confocal microscope. The ratio of fluorescence excited at 405 nm and 488 nm (R405/488) was used to evaluate the intracellular content of NADPH. R488/405 was used to evaluated the content of $H_2O_2$.

### De novo fatty acid synthesis

The cells were set up in 35-mm dishes. On the indicate days of differentiation, control, FASN-overexpressing or MED20-deficient adipocytes were incubated with DMEM containing uniformly [13]C-labeled glucose (25 mM) for 24 h. Cells were harvested for lipid extraction and saponification as previously described[54]. Briefly, cells were extracted in 1 ml 50% methanol containing 0.1 M of HCl (pre-chilled to −20 °C), and the resulting products were scraped into a glass vial. Chloroform (0.5 ml) was added, and the mixture was vortexed and centrifuged at 3000 rpm for 15 min. The organic phase was dried under nitrogen gas, and reconstituted into 90% methanol solution containing 0.3 M KOH. The mixture was incubated at 80 °C for 1 h, acidified, extracted with hexane. Dried samples were resuspended in 150 µl dichloromethane: methanol and subjected to LC/MS analysis.

### Gene expression analysis

Total RNA was isolated on day 12 of differentiation from control and MED20-deficient adipocytes, or adipose tissues from control and *Med20-AKO* mice. RNAs from 3 samples were pooled and subjected to RNA-Seq analysis as previously described[49]. Genes with a FPKM no less than 1 were included in the analysis. Quantitative real-time PCR (qRT-PCR) measurements were performed as described[55]. The primers are listed in Supplementary Table 1. All reactions were done in triplicates. The relative amount of each mRNA was calculated by using the comparative threshold cycle ($C_T$) method. Cyclophilin or 36B4 mRNA was used as the invariant control.

### Western blot and antibodies

Total proteins of adipose tissues were extracted and western blot was carried out as previously described[51]. The following antibodies were used: anti-β-Actin (1:1000, Protein Tech, 60008-1-Ig), anti-MED20 (1;1000, Protein Tech, 17598-1-AP), anti-GAPDH (1:5000, CST, 5174 s), anti-MLKL (1:1000, Abcam, ab184718), anti-P-MLKL (1:1000; Abcam, ab187091), anti-Cleaved Caspase-3 (1:1000, CST, 4190), anti-Acetyl-CoA Carboxylase1 (1:1000, CST, 4190), anti-ACLY (1:1000, CST, 13390), anti-RIPK3 (1:1000, ABclonal, A5431), anti-RIPK1 (1:1000, ABclonal, A7414), anti-P-RIPK3 (1:1000, Abcam, ab205421), anti-P-RIPK1 (1:1000, ABclonal, AP1230), anti-C/EBPα (1:1000, CST, 8178 S), anti-PPARγ (1:1000, Santa Cruz, sc-7273), anti-CD36 (1:1000, Sino Biological Inc, 80263-T48), anti-Perilipin 1 (1:1000, CST, 9349 s) and anti-FASN(1:1000, CST, 3180).

### Statistics and Reproducibility

All the statistical analysis was performed by student's two-tailed paired *t*-test using EXCEL2010. All the values represent mean ± s.e.m. All the statistical details of the experiments can be found in the figure legends, including exact number of cells or mice. No data were excluded from any of the experiments. All the experiments were repeated at least twice.

### Reporting summary

Further information on research design is available in the Nature Portfolio Reporting Summary linked to this article.

## Data availability

All data generated or analyzed during this study are included in this published article and its supplementary information files. The RNA-Seq data generated in this study have been deposited in the GEO database under accession code: GSE249808. Source data are provided with this paper.

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

## Acknowledgements

This work was supported by the National Key R&D Program of China (2020YFA0803601 to T.-J.Z.), the National Natural Science Foundation of China (32125022, 32230053 and 92157301 to T.-J.Z.; 32301083 to L.W.; 82270928 to C.-Q.L.; 32101046 and 32371364 to X.W.; 32201063 to W.-S.T.), the China Postdoctoral Science Foundation (2023M730691 to W.-S.T.), Excellent Youth Project of Natural Science Research of Anhui Education Department (2022AH030079 to X.W.), and Shanghai Basic Research Field Project "Science and Technology Innovation Action Plan" (21JC1400400 to P.L.).

## Author contributions

L.W., W.-S.T., X.W., N.-N. H., Y.T. and K.-Z.L. performed the experiments. C.-Q.L., Y.-Y.G., W.J., Y.Li and H.-W.Z. performed clinical study of the patient with acquired lipodystrophy. S.-N.L. and Y.-Z.Z. helped to perform the analysis of NADPH and H2O2 sensors. H.H. helped to analyze de novo fatty acid synthesis. K.Y., Z.-P.H., L.C., Y.Lu, H.Y. and X.D. provided materials and expertise. P.L. and T.-J.Z. designed the experiments, analyzed the data and wrote the manuscript.

## Competing interests

The authors declare no competing interests.
