## [Peer Review File · Nature Communications]

Surplus Fatty Acid Synthesis Increases Oxidative Stress in Adipocytes and Induces LipodystrophyREVIEWERS' COMMENTS:

Reviewer #1 (Remarks to the Author):

The authors of the present study report that excess fatty acid synthesis in the adipocytes is detrimental and that it induces cell death via necroptosis. Also, the increased FFA synthesis results in a decrease in NADPH, a cofactor required for FFA synthesis and increases ROS. The author's further show that this is mediated by MED20, a negative regulator of FFA synthesis and they generated an AT-Med20-KO mouse line that develops lipodystrophy reversed by scavenging ROS. They also tested this in a murine model of HIV-associated lipodystrophy and in a human lipodystrophic subject.

When I read the abstract before agreeing to review this, the study seems interesting.

However, upon reading the manuscript my enthusiasm evaporated. First pass, this seems to be a logical progression of their study, on deep diving it falls apart, several questions emerge:

1. The authors measured NADPH in the whole cell, not the subcellular region where FFA is synthesized. It is well known that NADP/NADPH is highly compartmentalized authors may like to read (PMID: 15774561, PMID: 24882210).
2. The authors show only a mild activation of FASN (2 fold), whereas there are several other tissue/cells where de novo lipogenesis (FASN) is increased by several fold more and still no cell death? Hepatocytes comes to mind. How do the authors reconcile this other than the effect is adipocyte-specific?
3. A negative regulator of FASN, MED20, increases the FFA "by how much" is not mentioned?
4. The authors show necroptosis by immunohistochemical method, by decrease in cell number and release of LDH. Both are considered non-specific methods for cell death. Although they do show increased level of phospho-MLKL (for necroptosis), the reviewer is unfamiliar with this approach and authors provide no reference. Additionally, authors do not show unphospho-MLKL since it is the phospho-MLKL which increases necroptosis. MLKL is part of RIP1 and RIP3 pathway, which needs to be studied.
5. In reviewers view, cellular ROS is inadequately estimated: The authors should consult any recent review on how to estimate and report ROS (eg., PMID: 29739855).
6. The authors show the presence of macrophages in the WAT and BAT MED-AKO mice.

Earlier it was suggested that these very inflammatory markers also help generate/maintain adipose tissue. So not sure whether it is because of AT destruction or regeneration? (PMID: 24930973).

7. Treating murine model (HIV) and human subjects with ROS mopping agents helps alleviate the lipodystrophy: again, from the reviewer's perspective, this is a general feature. After all, ROS inhibitors are consumed regularly. Not mentioned: does the human subject have any genetic alteration that could/might explain these observations? In addition, the human subject lipodystrophic feature very subtle and reviewer is unable to distinguish between treated and untreated.

8. Is there any MED20 variant reported in the literature? This is a very common component of transcription and will result in wide spread lipodystrophy. Or this is a tissue specific, but how?

9. Likewise MED20-AKO mice should have additional tissue effect and plasma H₂O₂ is the net effect of all tissues combined

Fig 1 – Panel I, M. Perform additional statistics on control and untreated vs NEC-1, LIP-1, ZVAD. Looking at the bars, LIP-1 and ZVAD are insignificant.

MED20-AKO mice require tamoxifen for activation, which is an antiestrogen. Not sure if this will affect the adipose tissue in females. After all, female hormones should affect the adipose tissue. Male and female adipose tissue anatomical depots are slightly different.

Fig 6 – The authors provide stavudine in drinking water, which also means each mice gets a different dose of stavudine (depending on how much water the mice drink). The reviewer is not fond of this approach for providing drugs. Panel I(i) – regarding authors' immunostaining of iWAT for anti-perilipin and Galectin3. Not sure how perilipin confirms necroptosis. The reviewer sees no membrane rupture. Perilipin mostly suggests the presence of adipocytes/lipid droplets.

Suppl. Table 2 – the patient's lipodystrophic features presented in this table are unremarkable.

Quantification of intracellular NADPH and H₂O₂ were carried out in immortalized preadipocytes. Immortalized cells are like transformed cells (like cancer cells) known to have increased H₂O₂. This is an inappropriate cell line to measure H₂O₂.

Overall, the study is insufficiently executed which makes the data interpretation difficult for this reviewer. What will be the source of plasma H₂O₂? Most H₂O₂ are intracellular. Do the

authors suggest it is secreted into the blood? Alternatively, the source of plasma H₂O₂ is different from adipocytes/adipose tissue, but not mentioned in this study.

Overall, the reviewer is uncomfortable recommending its publication, however, editors can always over rule this and the reviewer will have no quarrel with them.

Reviewer #2 (Remarks to the Author):

The study performed by Weng et al., enriches our knowledge and is important to the field in two important aspects: 1) Increased fatty acid synthesis rates in adipocytes induce lipodystrophy by inducing ROS stress and necroptosis, that's why adipocytes suppress FASN in a MED20 dependent manner and 2) Reducing ROS by Glutathione reverses stavudine and environmental oxidative stress induced lipodystrophy in mice and in a human patient, respectively.

1) The title of the manuscript only covers the first aspect. Therefore, just as a suggestion, authors may want to rethink the title.

2) Authors previously showed that MED20 is essential for adipogenesis by bringing together C/EBPbeta and RNA polymerase (Cell Reports, 2021). In the current manuscript they provide evidence that MED20 also functions to prevent oxidative stress by reducing FASN expression in adipocytes and deleting MED20 in mature adipocytes causes ROS induced necroptosis. For cell culture experiments this conclusion is very well taken. In mice, however, a constitutive active AdipoQ-promoter driven Cre was used to delete MED20. Accordingly, MED20 was deleted already during adipose tissue development. To discriminate between potential differentiation defects and acquired ROS induced deteriorations, authors may prefer to use an inducible Cre mouse for deletion of MED20 in adult mice.

3) Authors showed that reducing ROS by BHA treatment largely (but not completely) protects from lipodystrophy and associated metabolic deteriorations in MED20 knock out mice. Interestingly preventing necroptosis using NEC-1 also improves the lipodystrophy phenotype and comorbidities. However, inhibition of necroptosis will presumably not resolve the cellular ROS stress induced by MED20 deficiency. Therefore, a burning question is, how do adipocytes deal with this ROS stress if they cannot die? Or do they die, just not by necroptosis but by apoptosis? Can the authors maybe comment on that?

4) In a different disease model of lipodystrophy, and completely independent of FASN activity, authors found that detoxifying ROS by Glutathione reverses stavudine induced redistribution of lipids from iWAT to the liver and provide an explanation and a possible treatment option for lipodystrophy in HIV patients treated with stavudine. Authors found that after 9 weeks of stavudine treatment mice showed selective loss of iWAT.

a) As mice are still on HFD, is it really loss of iWAT or rather no more increase in iWAT due to the treatment?

b) Do the authors have an explanation for why only iWAT and not other adipose tissue depots are affected by this type of disease?

c) Authors suggest that also stavudine induced lipodystrophy is caused by elevated ROS because GSH treatment ameliorated the phenotype. It would strengthen the conclusion if authors could show that there is increased ROS in adipose tissue and/or in adipocytes treated with Stavudine.

d) Authors conclude this section by summarizing that ROS production is the major cause of HIV-associated lipodystrophy. This seems a bit overinterpreted. I would rather conclude that ROS production is the major cause of lipodystrophy in Stavudine treated HIV patients.

5) Is there any specific reason why all tissue weights are presented as %? I generally recommend showing absolute values for tissue weights instead of showing iWAT% or tissue weight % (of body weight).

Reviewer #3 (Remarks to the Author):

The authors examined the physiological significance of fatty acid synthesis in adipocytes and its relationship with lipodystrophy, excessive fatty acid synthesis in adipocytes triggers necroptotic cell death and lipodystrophy. This study highlights the critical role of ROS and free fatty acid in adipocytes, providing new targets for the treatment of related diseases. It should be noted that this paper presents experimental results based on mouse models and human samples, but further research is needed to validate these findings and determine their applicability in clinical settings. Furthermore, the study has some limitations, such as a small sample size and the need for further exploration of the underlying mechanisms.

The data reported in this manuscript are very interesting and novel. However, a few minor

points that need to be further confirmed.

1. Whether surviving of the big lipid droplets can be abolished by NEC-1, BHA or GSH? In addition, Nec-1 also inhibits apoptosis and autophagy in other reference

“The necroptosis inhibitor NEC-1, but not apoptosis inhibitor ZVAD or the ferroptosis inhibitor LIP-1, largely prevented cell death of FASN-overexpressing adipocyte”

2. It would be beneficial to perform a quantitative analysis of the WB band in Fig. 1c, 1n, 4a, Fig.s2g, Fig. s3d.

3. In Fig. s4g-j, why are the time points set differently for BAT, iWAT, gWAT and Liver?

4. In Fig.1 overexpression of FASN increased p-MLKL, but it is necessary to verify through alternative methods that an excessive production of fatty acids causes lipodystrophy through necroptosis.

5. It would be better to observe necroptosis in animal study.

6. The symptoms mentioned in this article seem to be more similar to type 2 diabetes. Is there any animal study on type 2 diabetes?

7. It would be better to organized the abbreviations into a table format.

8. In the human study, is there a direct correlation between the factors causing this disease and long-term engagement in occupations involving regular exposure to oxidants? Are there any relevant references available for this?

Reviewer #1 :

The authors of the present study report that excess fatty acid synthesis in the adipocytes is detrimental and that it induces cell death via necroptosis. Also, the increased FFA synthesis results in a decrease in NADPH, a cofactor required for FFA synthesis and increases ROS. The author's further show that this is mediated by MED20, a negative regulator of FFA synthesis and they generated an AT-Med20-KO mouse line that develops lipodystrophy reversed by scavenging ROS. They also tested this in a murine model of HIV-associated lipodystrophy and in a human lipodystrophic subject.

When I read the abstract before agreeing to review this, the study seems interesting. However, upon reading the manuscript my enthusiasm evaporated. First pass, this seems to be a logical progression of their study, on deep diving it falls apart, several questions emerge:

1. The authors measured NADPH in the whole cell, not the subcellular region where FFA is synthesized. It is well known that NADP/NADPH is highly compartmentalized authors may like to read (PMID: 15774561, PMID: 24882210).

Response: Thank you. Actually, the fluorescent indicator of NADPH (iNAP1) used in our manuscript (PMID:28581494, PMID:30258175) is to indicate the level of NADPH in the cytosol, where the de novo fatty acids synthesis happens. We stated it more clearly in the revised manuscript.

2. The authors show only a mild activation of FASN (2 fold), whereas there are several other tissue/cells where de novo lipogenesis (FASN) is increased by several fold more and still no cell death? Hepatocytes comes to mind. How do the authors reconcile this other than the effect is adipocyte-specific?

Response: Thank you for raising the point. This is a very interesting question. As we pointed out in the "Introduction", liver is the primary site for de novo fatty acid synthesis, whereas the fatty acid synthesis rate in other tissues including adipose tissues is neglectable (PMID: 10918543). De novo fatty acid synthesis is highly demanding on NADPH. For examples, the synthesis of 1 molecule of palmitate (C16:0) from acetyl-CoA consumes 14 molecules of NADPH. As reported in the literature (PMID: 33670839, PMID: 19803748, PMID:18796312), liver has the highest GSH in the body. The GSH content in hepatocytes can go up to 10 mM, while most other cells have a GSH content around 1-2 mM. This could be the reason how hepatocytes combat with the oxidative stress induced by de novo fatty acids synthesis. We added the following paragraph in the "Discussion" section.

It should be noted that liver is the primary site for de novo fatty acid synthesis; however, the hepatocytes, unlike the adipocytes, are able to manage the ROS induced by de novo fatty acid synthesis. A possible explanation is that hepatocytes have the highest GSH content in the body, which can go up to 10 mM, whereas other cell types have a GSH concentration around 1-2 mM (PMID: 33670839, PMID: 19803748,

PMID:18796312). The high GSH content might help to manage the oxidative stress caused by high rates of de novo fatty acid synthesis.

3. A negative regulator of FASN, MED20, increases the FFA “by how much” is not mentioned?

Response: Thanks. We actually included the data in the previously submitted manuscript (now Fig. 3i). There was a 2- to 3-fold increase in the newly synthesized C16:0, C16:1, C18:0 and C18:1 in the *Med20*-depleted adipocytes. We have made the changes in the revised manuscript.

Fig.3i. De novo fatty acid synthesis of control and MED20-deficient adipocytes.

Primary SVFs were isolated from *Rosa-Cre^{ERT2};Med20^{fl/fl}* mice, differentiated into mature adipocytes, and induced with 4-OHT to delete *Med20* on day 6. on day 9, cells were switched to medium containing uniformly ¹³C-labeled glucose (25 mM) for 24 h. Incorporation of ¹³C into the indicated fatty acids were analyzed.

4. The authors show necroptosis by immunohistochemical method, by decrease in cell number and release of LDH. Both are considered non-specific methods for cell death. Although they do show increased level of phospho-MLKL (for necroptosis), the reviewer is unfamiliar with this approach and authors provide no reference. Additionally, authors do not show unphospho-MLKL since it is the phospho-MLKL which increases necroptosis. MLKL is part of RIP1 and RIP3 pathway, which needs to be studied.

Response: Thank you for your question. Phospho-MLKL has been well documented and widely accepted as an executioner of necroptosis (PMID:29358703, PMID:32296174). After initiation of necroptosis, p-MLKL homo-oligomerizes and translocates to the plasma membrane to induce necroptosis. We have now included the reference in the revised manuscript.

In terms of the unphospho-MLKL, we actually showed total level of MLKL in the previous version of the manuscript (Fig. 1n), which did not show much difference between control and FASN^{oe} cells.

In the revised manuscript, we performed new experiments and detected phosphorylated and total protein levels of MLKL, RIPK1, and RIPK3. The results show that the phosphorylated forms of RIPK1 (1.4-fold), RIPK3 (2.6-fold) and MLKL (3.3-fold) are all significantly increased in FASN-overexpressing adipocytes (Fig. 1o;

Supplementary Fig. 1e).

To further strengthen the point, we performed the same experiment in control and *Med20*-AKO adipocytes and found that knocking out *Med20* caused an increase in the p-RIPK1, p-RIPK3 and p-MLKL (Supplementary Fig. 2m, n), which further confirms the necroptosis of adipocytes.

Fig.1o and Supplementary Fig. 1e. Protein level of necroptosis markers in control and FASN-overexpressing adipocytes.

On day 14 of differentiation, control and FASN-overexpressing adipocytes were subjected to western blot using indicated antibodies (Fig. 1o) and quantitative analysis of protein level (Supplementary Fig. 1e).

Supplementary Fig.2m and Supplementary Fig. 2n. protein level of necroptosis markers in Control and MED20-depleted (4-OHT) adipocytes.

On day 10 of differentiation, Control and MED20-depleted (4-OHT) adipocytes subjected to western blot using indicated antibodies (Supplementary Fig. 2m) and quantification analysis of protein level (Supplementary Fig. 2n).

5. In reviewers view, cellular ROS is inadequately estimated: The authors should consult any recent review on how to estimate and report ROS (eg., PMID: 29739855). Response: Thank you for your advice. In the current manuscript, we mainly used two ways to analyze and quantified the cellular ROS level. First, we used the genetical fluorescent sensor to monitor H_2O_2 level. Second, we used DCFDA, a fluorescent probe, to image and quantify ROS content.

After reading the reference suggested by the reviewer and other references, we used CellROX, which is a more specific fluorescent dye for ROS, to quantify the cellular ROS levels that were previously done using DCFDA. Overall, the results are consistent with those obtained from using DCFDA. We have now replaced all the DCFDA data with the CellROX data in Fig. 2e, f, Supplementary Fig. 2k, Fig. 3j, Supplementary Fig. 6a and Fig. 6c.

Fig. 2e, f. Analysis of ROS level in control and FASN-overexpressing adipocytes. On day 12 of differentiation, control and FASN-overexpressing adipocytes were subjected to CellROX (2.5 μ M) staining (Fig. 2e) and data were processed (Fig. 2f). each value represents mean \pm s.e.m. from 70 cells.

Supplementary Fig. 2k and Fig. 3j. Analysis of ROS level in Control and MED20-depleted (4-OHT) adipocytes with or without C75.

On day 12 of differentiation, Control and MED20-depleted (4-OHT) adipocytes subjected to CellROX staining (Supplementary Fig. 2k) and data were processed (Fig. 3j). each value represents mean \pm s.e.m. from 70 cells.

Supplementary Fig. 6a and Fig. 6c. Analysis of ROS level in adipocytes treated with stavudine with or without GSH.

On day 4 of treatment, cells were treated with CellROX and harvested for ROS quantification (Supplementary Fig. 6a) and data were processed (Fig. 6c). each value

represents mean \pm s.e.m. from 70 cells.

6. The authors show the presence of macrophages in the WAT and BAT MED-AKO mice. Earlier it was suggested that these very inflammatory markers also help generate/maintain adipose tissue. So not sure whether it is because of AT destruction or regeneration? (PMID: 24930973).

Response: Thank you for raising the point. Based on our findings and the general view on macrophages in adipose tissues (PMID: 33561645), we speculate that the macrophages are recruited by dead adipocytes. The major function of the macrophages is to clean the dead cells, but they might also help to regenerate or maintain the adipose tissues (PMID: 24930973). We included the following paragraph in the "Discussion" section to discuss the potential role of macrophages.

Macrophages play a pivotal role in the homeostasis of adipose tissues (PMID: 33561645). In both WAT and BAT, we observed large amounts of macrophage infiltration and elevation of inflammatory markers in *Med20-AKO* mice. The inflammation was dramatically improved after treating the mice with ROS scavenging reagent or necroptosis inhibitors. Therefore, we speculate that the macrophages are recruited by the dead adipocytes. In addition to clean the dead adipocytes, the macrophages might also help to regenerate the adipose tissues (PMID: 24930973).

7. Treating murine model (HIV) and human subjects with ROS mopping agents helps alleviate the lipodystrophy: again, from the reviewer's perspective, this is a general feature. After all, ROS inhibitors are consumed regularly. Not mentioned: does the human subject have any genetic alteration that could/might explain these observations? In addition, the human subject lipodystrophic feature very subtle and reviewer is unable to distinguish between treated and untreated.

Response: Thank you for the comments.

First, when we first diagnosed the patient, we thought there might be some genetic alterations that caused her symptoms. The patient is the only one that developed lipodystrophy in her family. We performed whole-genome exon sequencing of the patient, her brother and parents, but did not find any known mutations that might lead to lipodystrophy (Figure 1 for reviewer). Considering that she did not develop lipodystrophy until in her forties, we diagnosed her with acquired lipodystrophy.

In terms of improvement with her lipodystrophic feature, we presented her data in Figure 7. After 6 months of GSH treatment, while the increase in her adipose tissue might seem to be subtle, liver steatosis grade of the patient was dramatically improved based on the CAP score (Fig. 7e) and liver fat content measurement by MRI-PDFF (Fig. 7f). Based on the ALT and AST levels (Fig. 7h), her liver function return to normal. Currently, these clinical outcomes indicate that antioxidants treatment slowed down the disease process and may bring more benefits in the long term.

Figure 1 for reviewer

Figure 1 for reviewer. Whole-exon analysis of the patient and her family members. We performed whole-exon sequencing of the patient and her family members. The mutations inherited from her father and her mother were colored in blue and red, respectively.

8. Is there any MED20 variant reported in the literature? This is a very common component of transcription and will result in wide spread lipodystrophy. Or this is a tissue specific, but how?

Response: Thank you. We searched the literature and found that there is a homozygous mutation in MED20 (p. Gly114Ala) reported in 2015 (PMID:25446406). The mutation-bearing siblings were diagnosed with infantile-onset spasticity and childhood-onset dystonia, progressive basal ganglia degeneration and the brain atrophy (PMID:25446406). It was not documented about whether the patients developed lipodystrophy. We have added it the "Discussion" section.

As the reviewer pointed out, MED20 is a common component of the Mediator complex; however, MED20 has been shown to be a non-essential subunit of the Mediator complex (PMID: 31402173; 22341791). In the literature, little is known about the mechanism of MED20. We have previously shown that MED20 acts as a functional bridge between C/EBP β and RNA polymerase II to control the transcription of C/EBP α and PPAR γ , thereby regulating adipogenesis (PMID:34233190). Here, we show that MED20 controls the transcription of *Snai1* and *Snai2* to inhibit the transcription of *Fasn*. Taken together, we speculate that in different contexts, MED20 might interact with different transcription factors to control the transcription of downstream genes. We added the following paragraph in the "Discussion" section of the revised manuscript.

Although MED20 is a common component of the Mediator complex, it has been shown to be a non-essential subunit of the Mediator complex (PMID: 31402173; 22341791). We have previously shown that MED20 acts as a functional bridge between C/EBP β and RNA polymerase II to control the transcription of C/EBP α and

PPAR γ , thereby regulating adipogenesis (PMID:34233190). Here, we show that MED20 controls the transcription of *Snai1* and *Snai2* to inhibit the transcription of *Fasn*. Taken together, we speculate that in different contexts, MED20 might interact with different transcription factors to control the transcription of downstream genes.

9. Likewise MED20-AKO mice should have additional tissue effect and plasma H₂O₂ is the net effect of all tissues combined

Response:

Response: *Med20-AKO* mice are adipose-specific *Med20* knockout mice. Thus, the plasma H₂O₂ should be mainly contributed by dysfunction in adipose tissues.

Minor points

1. Fig 1 – Panel I, M. Perform additional statistics on control and untreated vs NEC-1, LIP-1, ZVAD. Looking at the bars, LIP-1 and ZVAD are insignificant.

Response: Thank you for your suggestion. We performed the statistics and found that the differences are insignificant. We therefore did not label the p-values on the figure.

2. MED20-AKO mice require tamoxifen for activation, which is an antiestrogen. Not sure if this will affect the adipose tissue in females. After all, female hormones should affect the adipose tissue. Male and female adipose tissue anatomical depots are slightly different.

Response: Thanks. As mentioned earlier, *Med20-AKO* mice are adipose-specific *Med20* knockout mice, which do not need tamoxifen to induce deletion of MED20.

We have also examined female mice and found that the female *Med20-AKO* mice also developed lipodystrophy. We have now included the data in Figure 2 for the Reviewer.

Figure 2 for reviewer

Figure 2 for reviewer. Adipocyte-specific knockout of *Med20* causes lipodystrophy in female mice.

a-j *Med20^{ff}* mice were crossed with AdipoQ-Cre mice to generate adipose tissue-specific *Med20* knockout mice (*Med20-AKO*). We choose female mice for the experiments. a body weight of female mice was monitored every week. Glucose (b)

and insulin (c) tolerance tests were performed on week 11 and week 12, respectively. Body composition (d) was analyzed on week 13. After that, mice were euthanized, and tissue were collected and weighed (e). Represent images of iWAT, gWAT, BAT and liver were shown (f). Plasma insulin (g, left), plasma leptin (g, right), plasma triglyceride (h, left) and liver triglyceride (h, right) were measured. iWAT, gWAT and BAT were fixed, sliced and subjected to H&E (i) and immunofluorescence using anti-Perilipin and anti-Galectin3 antibodies (j).

3. Fig 6 – The authors provide stavudine in drinking water, which also means each mice gets a different dose of stavudine (depending on how much water the mice drink). The reviewer is not fond of this approach for providing drugs. Panel I(i) – regarding authors' immunostaining of iWAT for anti-perilipin and Galectin3. Not sure how perilipin confirms necroptosis. The reviewer sees no membrane rupture. Perilipin mostly suggests the presence of adipocytes/lipid droplets.

Response: Thank you for raising the point. In our preliminary studies, administration of stavudine by both IP injection (twice daily) and supplementation in drinking water could efficiently induced the partial lipodystrophy model. As we need to observe iWAT in the experiment, IP injection twice daily would damage the morphology of the iWAT, we therefore used drinking water to deliver stavudine in the experiment. We also reviewed other published papers and found that stavudine is mainly administrated in water in the literatures (PMID:17591029, PMID:11303038, PMID:15535418). The amounts of stavudine added was calculated on the basis of a daily liquid consumption of 4 mL per mouse.

Regarding immunostaining using anti-perilipin and Galectin3, we apologize for not making it clear. As the reviewer pointed out, the use of anti-perilipin was to indicate the presence of adipocytes/lipid droplet. We used anti-Galectin3 to indicate macrophage infiltration. As shown in Fig. 6i, we observed much bigger adipocytes with increased macrophage infiltration in the stavudine treated iWAT, which was largely reversed by GSH treatment.

4. Suppl. Table 2 – the patient's lipodystrophic features presented in this table are unremarkable.

Response: Thank you for your question. In this table, we listed all the parameters we measured, including those that did not show much difference. Among them, ALT, AST and CAP showed big improvement after the treatment.

5. Quantification of intracellular NADPH and H₂O₂ were carried out in immortalized preadipocytes. Immortalized cells are like transformed cells (like cancer cells) known to have increased H₂O₂. This is an inappropriate cell line to measure H₂O₂.

Response: Thank you for raising the point. Ideally, it would be better to used freshly prepared adipocytes for the experiment. However, practically it is very challenging. we need to introduce the genetic fluorescent sensor proteins into the preadipocytes, select for infected cells, and differentiate into mature adipocytes, which takes at least two weeks. As primary stromal vascular fractions are not able to survive that long, we thus

used immortalized preadipocytes for the experiment.

As we could rescue the lipodystrophic phenotype in *Med20-AKO* mice by GSH or BHA, we hope to convince you that the results we obtained from immortalized preadipocytes are valid and could mimic the results in primary preadipocytes.

6. Overall, the study is insufficiently executed which makes the data interpretation difficult for this reviewer. What will be the source of plasma H₂O₂? Most H₂O₂ are intracellular. Do the authors suggest it is secreted into the blood? Alternatively, the source of plasma H₂O₂ is different from adipocytes/adipose tissue, but not mentioned in this study. Overall, the reviewer is uncomfortable recommending its publication, however, editors can always over rule this and the reviewer will have no quarrel with them.

Response: Thank you for raising the point. As *Med20-AKO* mice are adipose-specific knockout mice, we speculate that the major source of increase in plasma H₂O₂ is from adipose tissues. As elevated *de novo* fatty acid synthesis increases cellular ROS levels and causes cell death, we thus speculate that the plasma H₂O₂ is released from the dead adipocytes.

Reviewer #2

The study performed by Weng et al., enriches our knowledge and is important to the field in two important aspects: 1) Increased fatty acid synthesis rates in adipocytes induce lipodystrophy by inducing ROS stress and necroptosis, that's why adipocytes suppress FASN in a MED20 dependent manner and 2) Reducing ROS by Glutathione reverses stavudine and environmental oxidative stress induced lipodystrophy in mice and in a human patient, respectively.

1. The title of the manuscript only covers the first aspect. Therefore, just as a suggestion, authors may want to rethink the title.

Response: Thank you for your advice. We have changed the title to "Surplus Fatty Acid Synthesis Increases Oxidative Stress in Adipocytes and Induces Lipodystrophy".

2. Authors previously showed that MED20 is essential for adipogenesis by bringing together C/EBPbeta and RNA polymerase (Cell Reports, 2021). In the current manuscript they provide evidence that MED20 also functions to prevent oxidative stress by reducing FASN expression in adipocytes and deleting MED20 in mature adipocytes causes ROS induced necroptosis. For cell culture experiments this conclusion is very well taken. In mice, however, a constitutive active AdipoQ-promoter driven Cre was used to delete MED20. Accordingly, MED20 was deleted already during adipose tissue development. To discriminate between potential differentiation defects

and acquired ROS induced deteriorations, authors may prefer to use an inducible Cre mouse for deletion of MED20 in adult mice.

Response: Thank you for raising the point. *AdipoQ* is a specific marker of mature adipocytes, not preadipocytes. In 2011, the *AdipoQ*-Cre mice was developed to specifically manipulate gene expression in mature adipocytes (PMID: 21356515). In 2014, the *AdipoQ*-Cre was shown to be highly efficient and specific to mature adipocytes, and cannot induce gene deletion in preadipocytes (PMID: 25068087). So far, it has been used in more than 200 studies (PMID: 33371773).

To clarify the point, we isolated primary SVFs, and found that the protein level of MED20 in the preadipocytes of *Med20-AKO* mice did not differ from that of control mice (Supplementary Fig. 3g, h). However, after 9 days of differentiation, the protein level of MED20 was dramatically decreased in *Med20-AKO* adipocytes (Supplementary Fig. 3g, h). Thus, the *adipoQ*-Cre did not delete *Med20* in preadipocytes, but in adipocytes. Furthermore, on day 9 of differentiation, the *Med20-AKO* cells fully differentiated into mature adipocytes (Supplementary Fig. 3i-k), and the marker proteins of mature adipocytes, C/EBP α , PPAR γ and Perilipin1, did not show any difference between control and *Med20-AKO* adipocytes (Supplementary Fig. 3g, h). On day 12 of differentiation, we noticed cell death of adipocytes from *Med20-AKO* mice. By day 15 and day 18, the majority of the *Med20*-deficient adipocytes died, while the control adipocytes survived well (Supplementary Fig. 3i-k). From these above, we conclude that the lipodystrophy of the *Med20-AKO* mice is not due to the deletion of *Med20* in preadipocytes.

Actually, we tried to inducibly knock out *Med20* in *Rosa-Cre^{ERT2}; Med20^{ff}* mice by tamoxifen; however, these mice died after two days of tamoxifen treatment so that we cannot use this animal model. As we currently do not have the *AdipoQ-Cre^{ERT2}* mice, it might take at least 9 months before we can get enough *AdipoQ-Cre^{ERT2}; Med20^{ff}* mice for the study. Considering that others and we have demonstrated that the *AdipoQ*-Cre does not target preadipocytes, we hope the reviewer can agree with us not to generate the *AdipoQ-Cre^{ERT2}; Med20^{ff}* mice for the study.

Supplementary
Figure 3g

Supplementary
Figure 3h

Supplementary
Figure 3i

Supplementary
Figure 3j,k

Supplementary Figure. 3g-k. The *AdipoQ-Cre* does not delete *Med20* in the preadipocytes.

Primary SVFs were isolated from *MED20^{ff}* or *MED20-AKO* mice and differentiated. Cells were harvested on day 0 or day 9 then subjected to western blot using indicated antibodies (Supplementary Fig. 3g) and quantification of MED20 protein level on day 0 or day 9 (Supplementary Fig. 3g). Cells with different differentiation days are harvested for imaging under bright field (Supplementary Fig. 3i), quantification of cell number (Supplementary Fig. 3j) and released LDH (Supplementary Fig. 3k).

3. Authors showed that reducing ROS by BHA treatment largely (but not completely) protects from lipodystrophy and associated metabolic deteriorations in MED20 knock out mice. Interestingly preventing necroptosis using NEC-1 also improves the lipodystrophy phenotype and comorbidities. However, inhibition of necroptosis will presumably not resolve the cellular ROS stress induced by MED20 deficiency. Therefore, a burning question is, how do adipocytes deal with this ROS stress if they cannot die? Or do they die, just not by necroptosis but by apoptosis? Can the authors maybe comment on that?

Response: Thank you. This is a great question. In supplementary Figure 5, we show that inhibition of necroptosis largely restored the fat mass of *Med20-AKO* mice. The surviving adipocytes were much bigger in size and the inflammation markers were decreased in NEC-1-treated adipose tissues. Therefore, inhibition of necroptosis likely

does not cause other types of cell death of the adipocytes, at least in the 8-week period of the experiment. As the reviewer pointed out, inhibition of necroptosis will presumably not resolve the cellular ROS stress induced by MED20 deficiency. Currently, we do not have a clear answer of how these adipocytes live with the ROS stress. Presumably, the adipocytes might develop a compensatory mechanism to resolve the ROS stress, or they might eventually die when the ROS levels accumulates to a certain level. We added the following paragraph in the “Discussion” section in the revised manuscript.

Interestingly preventing necroptosis using NEC-1 also improves the lipodystrophy phenotype and comorbidities; however, inhibition of necroptosis will presumably not resolve the cellular ROS stress induced by MED20 deficiency. We noticed that the surviving adipocytes were much bigger in size and the inflammation markers were decreased in NEC-1-treated adipose tissues. Therefore, inhibition of necroptosis likely does not cause other types of cell death of the adipocytes, at least in the 8-week period of the experiment. Currently, we do not have a clear answer about how these adipocytes live with the ROS stress. Presumably, the adipocytes might develop a compensatory mechanism to resolve the ROS stress, or they might eventually die when the ROS levels accumulates to a certain level.

4. In a different disease model of lipodystrophy, and completely independent of FASN activity, authors found that detoxifying ROS by Glutathione reverses stavudine induced redistribution of lipids from iWAT to the liver and provide an explanation and a possible treatment option for lipodystrophy in HIV patients treated with stavudine. Authors found that after 9 weeks of stavudine treatment mice showed selective loss of iWAT.

a) As mice are still on HFD, is it really loss of iWAT or rather no more increase in iWAT due to the treatment?

Response: Thank you for the question. In Fig. 6i, we examined stavudine-treated iWAT by H&E staining. We noticed that macrophage infiltration was dramatically more in stavudine-treated iWAT than that in control iWAT, and such a phenomenon was rescued by GSH treatment. Also, we noticed that the surviving adipocytes in stavudine-treated iWAT are much bigger in size. Considering that the sizes of iWAT is smaller in stavudine-treated mice (Fig. 6g, h), the number of adipocytes should be less in stavudine-treated iWAT.

Together with the data that stavudine treatment caused cell death in the cell culture experiment (Fig. 6a, b), we conclude that the loss of iWAT in stavudine treatment is due to cell death of the adipocytes.

b) Do the authors have an explanation for why only iWAT and not other adipose tissue depots are affected by this type of disease?

Response: Thank you for raising the point. It has been well documented that in HIV-associated lipodystrophy, nucleoside reverse transcriptase inhibitors (NRTIS) caused selective loss of subcutaneous fat; however, the underlying mechanism remains unknown. Presumably, stavudine has a specific target in iWAT that will selectively cause cell death of adipocytes in iWAT.

c) Authors suggest that also stavudine induced lipodystrophy is caused by elevated ROS because GSH treatment ameliorated the phenotype. It would strengthen the conclusion if authors could show that there is increased ROS in adipose tissue and/or in adipocytes treated with Stavudine.

Response: Thank you. We actually included the data in Fig. 6c of the previous version of the manuscript. Indeed, stavudine treatment caused an increase in ROS level (indicated by DCFDA), which was normalized by GSH treatment. In the revised manuscript, we also included the data obtained from using CellROX, another ROS indicator, and the conclusions are the same.

d) Authors conclude this section by summarizing that ROS production is the major cause of HIV-associated lipodystrophy. This seems a bit overinterpreted. I would rather conclude that ROS production is the major cause of lipodystrophy in Stavudine treated HIV patients.

Response: Thank you for your advice. We made the changes to the text as you suggested.

5. Is there any specific reason why all tissue weights are presented as %? I generally recommend showing absolute values for tissue weights instead of showing iWAT% or tissue weight % (of body weight).

Response: Thanks. This is the way we usually present our data. It happened to us before that when mice have developmental defect or malnutrition, their body and tissue weights would all decreased, but in proportion. In those cases, the decrease in adipose tissue is non-autonomous. We therefore usually present tissue weight as a percentage of body weight to rule out the development defect or malnutrition. If the reviewer thinks that showing the absolute weight might a better way to present the data, we can make the changes. In any case, the original data including tissue weights will be deposited in the source data file.

Reviewer #3

The authors examined the physiological significance of fatty acid synthesis in adipocytes and its relationship with lipodystrophy, excessive fatty acid synthesis in adipocytes triggers necroptotic cell death and lipodystrophy. This study highlights the critical role of ROS and free fatty acid in adipocytes, providing new targets for the treatment of related diseases.

It should be noted that this paper presents experimental results based on mouse models and human samples, but further research is needed to validate these findings and determine their applicability in clinical settings. Furthermore, the study has some limitations, such as a small sample size and the need for further exploration of the underlying mechanisms.

The data reported in this manuscript are very interesting and novel. However, a few minor points that need to be further confirmed.

1. Whether surviving of the big lipid droplets can be abolished by NEC-1, BHA or GSH?

In addition, Nec-1 also inhibits apoptosis and autophagy in other reference “The necroptosis inhibitor NEC-1, but not apoptosis inhibitor ZVAD or the ferroptosis inhibitor LIP-1, largely prevented cell death of FASN-overexpressing adipocyte”.

Response: Thank you for your question. We quantified the sizes of the lipid droplets in adipocytes treated with NEC-1, BHA or GSH. As you can see in the Figure 3 for reviewer, NEC-1 treatment slightly decreased the percentage of lipid droplets ranging from 500 to 1,000 μm^2 . Treatment with BHA or GSH slightly decreased the percentage of lipid droplets ranging from 1,000 to 3,000 μm^2 . Overall, the changes are very small. The results actually make sense, as none of the reagents above blocks de novo fatty acid synthesis in the adipocytes. Considering the limit of space, we did not include the figure in the revised manuscript.

In terms of NEC-1, thank you for your comment. To strengthen the point that FASN overexpressing adipocytes died of necroptosis. We performed western blot of p-RIPK1, p-RIPK3 and p-MLKL, all of which are specific markers of necroptosis. We found that these markers are significantly increased in FASN overexpressing adipocytes (Fig. 1o). To avoid any confusion, we made changes to our manuscript about NEC-1 as the followings.

Figure 3 for reviewer

Figure 3 for reviewer. Size analysis of lipid droplets in control and FASN-overexpressing adipocytes treated with NEC-1, BHA and GSH.

We analyzed the size of lipid droplet in control and FASN-overexpressing adipocytes treated with or without NEC-1(a) or BHA and GSH(b).

2. It would be beneficial to perform a quantitative analysis of the WB band in Fig. 1c, 1n, 4a, Fig.s2g, Fig. s3d.

Response: Thanks for your advice. We quantified the WB bands in these figures and included the data in the revised manuscript.

3. In Fig. s4g-j, why are the time points set differently for BAT, iWAT, gWAT and Liver?

Response: Thank you for raising the point. As shown in Fig. S4a-f, the difference in BAT, iWAT, gWAT and liver between control and *Med20-AKO* mice showed up at different time. We actually performed H&E staining of these tissues at all the time points. Space wise, we only presented the data that started to show no difference, then mild and big differences. Thus, the time points were different in each tissue.

4. In Fig.1 overexpression of FASN increased p-MLKL, but it is necessary to verify through alternative methods that an excessive production of fatty acids causes lipodystrophy through necroptosis.

Response: Thank you for the question. In Fig. 3 and Fig. S2, we showed that MED20 inhibits the transcription of *Fasn* through two transcriptional repressors, SNAIL and SLUG. We showed that knockdown of MED20, SNAIL or SLUG all increased the transcription of *Fasn* and induced necroptosis.

To further strength the point, we examined the levels of p-RIPK1, p-RIPK3 and p-MLKL in control and MED20-deficient adipocytes, and found that these necroptosis markers were dramatically increased in MED20-deficient adipocytes. These data are now included in Fig. S2m, n of the revised manuscript.

5. It would be better to observe necroptosis in animal study.

Response: Thank you for your suggestion. To check the necroptosis in the iWAT and BAT of *Med20^{ff}* and *Med20-AKO* mice, we performed immunohistochemistry using the p-MLKL antibody. We found that the p-MLKL signal was much stronger in the iWAT and BAT of *Med20-AKO* mice. We have included the data in Fig. 4k in the revised manuscript.

Figure. 4k

Figure 4k. Immunohistochemistry analysis of necroptosis in iWAT and BAT.

Immunohistochemistry was performed using anti-pMLKL antibody in iWAT (7 weeks old) and BAT (5 weeks old) from control and *Med20-AKO* mice.

6. The symptoms mentioned in this article seem to be more similar to type 2 diabetes. Is there any animal study on type 2 diabetes?

Response: Thanks. The symptoms mentioned in this article indeed resembled type 2 diabetes in glucose management, which is a common feature of lipodystrophy. We actually performed some animal studies about this in the manuscript. In Fig. 4, we performed glucose tolerance test and insulin tolerance test, and found that *Med20-AKO* mice showed defect in glucose clearance and insulin sensitivity. Furthermore, plasma insulin level was dramatically higher in *Med20-AKO* mice (Fig. 4i). From these results, we concluded that *Med20-AKO* mice developed symptoms resembling type 2 diabetes.

In addition, in Fig. 5 and S5 we performed glucose tolerance test and insulin tolerance test and showed that scavenging ROS or blocking necroptosis improved glucose clearance and insulin sensitivity of *Med20-AKO* mice. In Fig. 6, we showed

that stavudine treatment caused defect in glucose clearance and insulin sensitivity in WT mice.

7. It would be better to organized the abbreviations into a table format.

Response: Thanks for your advice. We re-organized the abbreviations and included them in Supplementary Table 3.

8. In the human study, is there a direct correlation between the factors causing this disease and long-term engagement in occupations involving regular exposure to oxidants? Are there any relevant references available for this?

Response: Thanks for your comment. It has been well documented that long-term engagement in occupations involving regular exposure to oxidants would cause an increase in plasma ROS levels (DOI: 10.1007/s13530-015-0216-2). However, in the population-based epidemiological studies, BMI and other common biological metabolic evaluations are used, without providing a precise evaluation of body fat distribution; therefore, the association has not been established between long-term engagement in occupations involving regular exposure to oxidants and lipodystrophy. In the future, it will be interesting to investigate the causal-effect study about oxidants exposure and fat mass and adipocyte function in humans. We included this paragraph in the "Discussion" section.

REVIEWER COMMENTS

Reviewer #1 (Remarks to the Author):

The revised manuscript is better than the original one however there remains a few previous comments not addressed adequately:

Regarding comment #1: The reviewer was looking more for a biochemical approach – like cell fractionation. The refs mentioned actually do not provide quantification of NADPH, only localization. For the reviewer, it does not cut it.

Regarding comment #2: The reviewer disagrees with the authors that the liver is the primary site of de-novo lipogenesis. It occurs only in pathological conditions, like fatty liver. An abnormal condition cannot be equated with normal situations. Furthermore, the authors argue about GSH content, but cite refs only. Did they ever even measure GSH in their study?

Regarding comment #7: The authors mention fatty liver in their patient; this can be due to liver dysfunction independent of adipose tissue. This possibility should have been ruled out. ROS is such an agent that helps ameliorate several conditions. The reviewer is not convinced with this data set.

Regarding comment #8: When there is a genetically-transmitted or de novo mutation, it occurs in all the cell types. So why will only one tissue be affected? What the authors should have performed is the expression level of MED20 (a tissue survey) and determine if MED20 expression (both mRNA, protein) to explain this.

Suppl. Table 2: lab findings are not enough. The authors could have provided quantification of various adipose tissue, using MRI, skin fold changes, etc. Partial lipodystrophy is always misdiagnosed. As one very famous investigator (cannot mention the name) once said, half the whole world is “partially lipodystrophic.” Authors did perform Exome sequencing, include this data set for all to see and read it should not be for “reviewers only”.

Minor points:

1. Fig 1 – Panel I, M... authors carried out the stats and mention they were NS. What will be a problem if the authors show those on the figure panel? How do a reader know that stats were done and likely NS and assume authors did not mention? Science should not be a guessing game.

As mentioned in my overall comment, the source of H₂O₂ in the plasma remains unanswered.

I am intrigued by the comment raised by reviewer #2 inducible MED20 deletion and the authors' response that the mice die after two days of TAM treatment. Why would the mice die when MED20 is only responsible for adipose tissue function? This supports this reviewer's argument that reduced adipose tissue and H₂O₂ are being modulated by tissue(s) other than adipose tissue.

Also, regarding the usage of adiponectin promoter to guide adipose tissue-specific expression: the authors mention that adiponectin is a specific marker of mature adipocytes and this will not affect pre-adipocyte development. Interesting. This reviewer searched the recent literature and found a paper (PMID: 37752957) where a similar adiponectin promoter was used embryonically for adipocyte regeneration. Although no embryonic histology is presented, it puts into question the authors' argument regarding the role of adiponectin promoter in pre-adipocytes vs mature adipocytes.

This reviewer suggests to remove all the human lipodystrophy data-sets as it is very weak and further address the still remaining issues. Editors can then decide how they wish to go forward with this study. This reviewer will not review this study further.

Reviewer #2 (Remarks to the Author):

In the revised manuscript submitted by Weng et al. and in the response letter the authors addressed all my questions and concerns. They performed additional experiments and adapted the manuscript. I believe this work enriches our basic knowledge about adipocyte metabolism and provides insights into how oxidative stress causes acquired lipodystrophy. Congratulations to the authors to their work!

Reviewer #3 (Remarks to the Author):

The authors addressed the reviewer' comments

Reviewer #1 (Remarks to the Author):

The revised manuscript is better than the original one however there remains a few previous comments not addressed adequately:

Regarding comment #1: The reviewer was looking more for a biochemical approach – like cell fractionation. The refs mentioned actually do not provide quantification of NADPH, only localization. For the reviewer, it does not cut it.

Response: In Fig. 2a and 2b, we have actually shown the localization and quantification of NADPH using the cytosolic iNAP fluorescent sensor. In theory, a biochemical approach like cell fraction will also be able to quantify the cytosol NADPH level. However, considering that NADPH is very unstable and hard to measure, cell fractionation might lead to the degradation of the cytosolic NADPH. Instead, the iNAP sensor can be used to measure the *in situ* NADPH level in the living cells, which we think is a better strategy.

Regarding comment #2: The reviewer disagrees with the authors that the liver is the primary site of de-novo lipogenesis. It occurs only in pathological conditions, like fatty liver. An abnormal condition cannot be equated with normal situations. Furthermore, the authors argue about GSH content, but cite refs only. Did they ever even measure GSH in their study?

Response: Sorry for the confusion. In this manuscript, we mainly focused on de novo fatty acid synthesis, not de novo lipogenesis which includes both fatty acid synthesis and the following conversion of fatty acids into triglycerides. As we stated in the introduction, even in the 1970s, it has been well documented that liver, not the adipose tissue, is the primary site for de novo fatty acid synthesis. We did not use “lipogenesis” in our manuscript.

As we focused on adipose tissue in the manuscript, we did not see a point in measuring the GSH content in liver, especially since it has been done before.

Regarding comment #7: The authors mention fatty liver in their patient; this can be due to liver dysfunction independent of adipose tissue. This possibility should have been ruled out. ROS is such an agent that helps ameliorate several conditions. The reviewer is not convinced with this data set.

Response: Based on our diagnosis, the patient developed acquired lipodystrophy. As fatty liver is a common feature of lipodystrophic patients, we thus believe that the fatty liver is due to the lack of adipose tissues.

As the reviewer pointed out, fatty liver could be due to many different reasons, including liver dysfunction; however, it is not possible to rule this out in a patient. Due to the potential ethic issues, it is not appropriate to test the hypothesis in the patient, especially knowing that she is lipodystrophic.

Regarding comment #8: When there is a genetically-transmitted or de novo mutation, it occurs in all the cell types. So why will only one tissue be affected? What the authors should have performed is the expression level of MED20 (a tissue survey) and

determine if MED20 expression (both mRNA, protein) to explains this.

Response: As we responded in the previous version, we used adipose-specific *Med20* knockout mice in the study, and that is why only adipose tissue is affected in the study.

Suppl. Table 2: lab findings are not enough. The authors could have provided quantification of various adipose tissue, using MRI, skin fold changes, etc. Partial lipodystrophy is always misdiagnosed. As one very famous investigator (cannot mention the name) once said, half the whole world is “partially lipodystrophic.” Authors did perform Exome sequencing, include this data set for all to see and read it should not be for “reviewers only”.

Response: In Supplemental Table 2, all the parameters were obtained from the medical equipment. This is how we usually diagnose patients. Indeed, an MRI scan to quantify various adipose tissues might help to further validate the results, but we could not ask the patient to quit GSH treatment and re-analyze her parameters. We hope you can understand that.

As you suggested, we have included the exosome sequencing result in the Supplemental Fig. 7c.

Minor points:

1. Fig 1 – Panel I, M... authors carried out the stats and mention they were NS. What will be a problem if the authors show those on the figure panel? How do a reader know that stats were done and likely NS and assume authors did not mention? Science should not be a guessing game.

Response: When we present the data, we usually only label the p values on those with statistical significance. It is also the style of the Nature Communications. Thus, it is about the scientific habit, not the guessing game.

As mentioned in my overall comment, the source of H₂O₂ in the plasma remains unanswered.

Response: We have shown that in Fig, 2c, the H₂O₂ production increased about 2.3-fold in FASN-overexpressing adipocytes. Also as mentioned in the previous version, *Med20-AKO* mice are adipose-specific *Med20* knockout mice. Thus, the plasma H₂O₂ should be mainly contributed by dysfunction in adipose tissues. As far as we know, it has not been reported about how to trace the source of H₂O₂. Based on our data, we believe that elevated plasma H₂O₂ was from *Med20-AKO* adipocytes.

I am intrigued by the comment raised by reviewer #2 inducible MED20 deletion and the authors’ response that the mice die after two days of TAM treatment. Why would the mice die when MED20 is only responsible for adipose tissue function? This supports this reviewer’s argument that reduced adipose tissue and H₂O₂ are being modulated by tissue(s) other than adipose tissue.

Also, regarding the usage of adiponectin promoter to guide adipose tissue-specific expression: the authors mention that adiponectin is a specific marker of mature adipocytes and this will not affect pre-adipocyte development. Interesting. This

reviewer searched the recent literature and found a paper (PMID: 37752957) where a similar adiponectin promoter was used embryonically for adipocyte regeneration. Although no embryonic histology is presented, it puts into question the authors' argument regarding the role of adiponectin promoter in pre-adipocytes vs mature adipocytes.

Response: In the previous response to the comment #2 of Reviewer #2, we have addressed the concerns by performing experiments to confirm that the adipoQ-Cre will not delete *Med20* in preadipocytes.

In terms of the paper mentioned by the reviewer (PMID: 37752957), the title is "Regulated adipose tissue-specific expression of human AGPAT2 in lipodystrophic *Agpat2*-null mice results in regeneration of adipose tissue". The authors actually used the adipoQ-Cre to re-express human AGPAT2 in the *Agpat2*^{-/-} mouse adipose tissues. They did not show any data that the adipoQ-Cre can target preadipocytes.

Again, with our data (Supplemental Fig. 3g-k) and the previous 200 papers using the AdipoQ-Cre, we hope to convince you that it is not an issue to use the AdipoQ-Cre for our study.

This reviewer suggests to remove all the human lipodystrophy data-sets as it is very weak and further address the still remaining issues. Editors can then decide how they wish to go forward with this study. This reviewer will not review this study further.

Response: As acquired lipodystrophy is a very rare disease, we only had one patient for the current study. We agree that the data from one patient might not represent a large population; however, with all the evidences we provided in the manuscript and previous studies using vitamin E to treat HIV-infected patients, we think that our findings might still shed light on the treatment of patients of acquired lipodystrophy. We therefore think we should keep the data in the manuscript.

We have added the following paragraph in the "Discussion" section to talk about the limit of the patient study and the potential of developing a therapeutic strategy for acquired lipodystrophy.

Notably, due to the rarity of acquired lipodystrophy, we only recruited one patient in the study. Thus, there might be some uncertainty to apply the observation in the current study to other patients with acquired lipodystrophy. However, based on the evidences we obtained from mouse studies and the metabolic benefits of anti-oxidant in HIV-infected patients receiving antiretroviral therapy³⁹, it is reasonable to deduce that increased oxidative stress might be a potential cause of acquired lipodystrophies. It is also promising that GSH might be a therapeutic treatment for patients with acquired lipodystrophy.

REVIEWERS' COMMENTS

Reviewer #4 (Remarks to the Author):

The authors of the manuscript entitled "Surplus Fatty Acid Synthesis increases Oxidative Stress in Adipocytes and induces Lipodystrophy" adequately addressed the suggestions and concerns of Reviewer 1.

Although, I agree with Reviewer 1 that the human data are weak, the authors explained in the discussion section the limitation for patient studies.

One minor comment:

Since there is a 45% decrease in adipocytes on day 15, it might be useful to mention the state of the adipocytes on day 9, when you start with the experiments for de novo fatty acid synthesis in Fig 1e-g (and for following experiments).

The statement that double knockdown of Snail and Slug further increased Fasn mRNA expression (Fig 3a) is not statistically proven. I would weaken or omit this statement, since you nicely show a synergistic effect in Fig 3 b/c.

Reviewer #4 (Remarks to the Author):

The authors of the manuscript entitled “Surplus Fatty Acid Synthesis increases Oxidative Stress in Adipocytes and induces Lipodystrophy” adequately addressed the suggestions and concerns of Reviewer 1.

Although, I agree with Reviewer 1 that the human data are weak, the authors explained in the discussion section the limitation for patient studies.

One minor comment:

Since there is a 45% decrease in adipocytes on day 15, it might be useful to mention the state of the adipocytes on day 9, when you start with the experiments for de novo fatty acid synthesis in Fig 1e-g (and for following experiments).

Response: Thank you very much. We added the following sentence to the revised manuscript: On the day of experiment, the control and FASN-overexpressing adipocytes did not show visible difference.

The statement that double knockdown of Snail and Slug further increased Fasn mRNA expression (Fig 3a) is not statistically proven. I would weaken or omit this statement, since you nicely show a synergistic effect in Fig 3 b/c.

Response: Thank you. As suggested, we omitted this statement in the revised manuscript.